# Investigation of spatial and temporal variability of river-ice phenology and thickness across Songhua River Basin, Northeast China

Qian Yang[1, 2], Kaishan Song[1, *], Xiaohua Hao[3], Zhidan Wen[2], Yue Tan[1], and Weibang Li[1]

[1] Jilin Jianzhu University, Xincheng Road 5088, Changchun 130118 China; E-Mail: jluyangqian10 @hotmail.com

[2] Northeast Institute of Geography and Agroecology, Chinese Academy of Sciences, Shengbei Street 4888, Changchun 130102 China; E-Mail: songks@neigae.ac.cn;

[3] Northwest Institute of Eco-Environment and Resources, Chinese Academy of Sciences, Donggang West Road 322, Lanzhou 730000, China; E-Mail: haoxh@lzb.ac.cn;

*Correspondence to*: Song K. S. (songks@neigae.ac.cn)

Abstract: The regional role and trends of freshwater ice are critical factors for aquatic ecosystems, climate variability, and human activities. The ice regime has been scarcely investigated in the Songhua River Basin of Northeast China. Using daily ice records of 156 hydrological stations across the region, we examined the spatial variability in the river ice phenology and river ice thickness from 2010 to 2015, and explored the role of snow depth and air temperature on the ice thickness. The river ice phenology showed a latitudinal distribution and a changing direction from southeast to northwest. We identified four spatial clusters based on Moran's I spatial autocorrelation, and results showed that the completely frozen duration with high values clustered in the Xiao Hinggan Mountains and that with low values clustered in the Changbai Mountains at the 95% confidence level. The maximum ice thickness over 125 cm was distributed along the ridge of Da Hinggan Mountain and Changbai Mountains, and the maximum ice thickness occurred most often in February and March. In three sub-basins of the Songhua River Basin, we developed six Bayesian regression models to predict ice thickness from air temperature and snow depth. The goodness of the fit ($R^2$) for these regression models ranged from 0.80 to 0.95, and the root mean square errors ranged from 0.08 to 0.18 meter. Results showed significant and positive correlations between snow cover and ice thickness when freshwater was

completely frozen. Ice thickness was influenced by cumulative air temperature of freezing through the heat loss of ice formation and decay, instead of just air temperature.

**Keywords.** River ice phenology, ice thickness, snow depth on ice, cumulative air temperature of freezing, Bayesian linear regression

## 1 Introduction

The freeze-thaw process of temperate lakes and rivers' surface ice plays a crucial role in the interactions among the climate system (Yang et al., 2020), the freshwater ecosystem (Kwok and Fahnestock, 1996) and the biological environment (Prowse and Beltaos, 2002). The presence of freshwater ice is closely associated with social and economic activities, such as from human-made structures, water transportation, and winter recreation (Lindenschmidt et al., 2017; Williams and Stefan, 2006). Ice cover on rivers and lakes exerts large forces due to thermal expansion and could cause extensive infrastructure losses to bridges, docks, and shorelines (Shuter et al., 2012). Ice cover on waterbodies also provides a natural barrier between the atmosphere and the water. Besides, ice cover also blocks the solar radiation, which is necessary for photosynthesis to provide enough dissolved oxygen for fish, thus posing a negative effect on freshwater ecosystems. In extreme cases, it can lead to the winter kill of fish (Hampton et al., 2017). Generally, the duration of freshwater ice has shown a declining trend, with later freeze-up and earlier break-up throughout the northern hemisphere. For example, the freeze-up has been occurring 0.57 days later per decade and the break-up 0.63 days earlier per decade during the periods of 1846-1995 (Beltaos and Prowse, 2009; Magnuson et al., 2000; Sharma et al., 2019). Despite the growing importance of river ice under global warming, very little work has been undertaken to explain the considerable variation of ice characteristics in Northeast China, where lakes and rivers are frozen for as long as five to six months a year. A robust and quantitative investigation on the variations of rive ice regime associated with changes in snow depth on ice and air temperature, are fundamental for understanding climate changes on regional scales.

The earliest ice record in the literatures dates back to 1840s throughout the northern hemisphere (Magnuson et al., 2000). Ice development and ice diversity scales have been regarded as sensitive climate indicators. Ice phenology and ice thickness have been studied

to obtain a deeper understanding of ice processes. The optical remote sensing data at medium and large scales are widely adopted for deriving ice phenology (Šmejkalová et al., 2016; Song et al., 2014). In contrast, microwave remote sensing are used to estimate ice thickness and snow depth over ice (Kang et al., 2014; Zhang et al., 2019). Wide-range satellites make it possible to link ice characteristic with climate indices, such as air

temperature (Yang et al., 2020) or large-scale teleconnections (Ionita et al., 2018). Still, their spatial resolutions are too coarse to detect ice thickness and the snow depth over ice at local scales accurately. For example, the microwave satellite data of AMSR-E have a spatial resolution of 25 km, but the largest width of the Nenjiang River only ranges from 1700 to 1800 meters. The spatial resolution limits the application of satellite observations

to inverse ice thickness precisely, let alone the snow depth.

In terms of point-based measurements, the most commonly used ground observations include the fixed-station observations, the ice charts, the volunteer monitoring and the field measurements (Duguay et al., 2015). Ground observations depend on the spatial

distribution and the representation, which are limited by the accessibility of surface-based networks. Various models, such as physically-based models (Park et al., 2016), linear regressions (Palecki and Barry, 1986; Williams and Stefan, 2006), logistic regressions (Yang et al., 2020) and artificial neural networks (Seidou et al., 2006; Zaier et al., 2010), have been developed to derive ice phenology and ice thickness. The physically-based

models mainly consider the energy exchange and physical changes of freshwater ice and require detailed information and data support, including hydrological, meteorological, hydraulic and morphological information (Rokaya et al., 2020). As the relevant information on local scales is more readily available, the physically-based models are more suitable for small watershed applications (e.g. within 100 km$^2$). On the other hand, empirical models

are more commonly adopted to predict changes in the ice regime from relatively limited climate data available over larger basins (Yang et al, 2020). Ice parameters, such as ice thickness, freeze-up and break-up dates, differ notably from point to point on a given river continuum (Pavelsky and Smith, 2004), and the uneven distribution of hydrological stations poses an obstacle for spatial investigation and modelling. Therefore, ssufficient historical

ice records are necessary to model the ice regime and validate the reliability of remote sensing data.

The ice cover of water bodies experiences three stages: the freeze-up, the ice growth, and the break-up (Duguay et al., 2015). The ice phenology, the ice thickness, and the ice composition change considerably in different stages. Although air temperature dramatically influences the freeze-thaw cycle of river ice dramatically, the effect of snow cover cannot be ignored. Generally, the effect of snow depth on the ice forming process is more vital than the impact of air temperature (Morris et al., 2005; Park et al., 2016). In contrast to these studies, Gao and Stefan (1997) found that the air temperature had a more substantial effect on the ice thickness formation than the snow depth. Furthermore, in situ observations at Russian river mouths, where ice thickness decreased, did not show any striking correlation between the ice thickness and the snow depth (Shiklomanov and Lammers, 2014). Previous studies have analysed the relationship in view of spatial distributions but ignored the frozen status of ice formation processes. The relative influence of snow depth and air temperature on the freshwater ice regimes in Northeast China calls for a detailed exploration.

To estimate the interaction between the ice regime and the climate systems, a comprehensive investigation and robust analysis on the ice regime are essential, which can provide relevant information for projecting future changes in the ice regime. The work is the first to present continuous river ice records of three sub-catchments of the Songhua River Basin from 2010 to 2015, and the study compares the spatial and temporal changes of ice phenology and ice thickness. The influence of snow cover and air temperature on the ice regime is quantitatively explored with the three sub-catchments considering the frozen status of the river ice.

## 2 Materials and methods

### 2.1 Study area

The Songhua River Basin is located in the middle area of Northeast China (Figure 1), which includes Liaoning Province, Jilin Province, Heilongjiang Province, and the eastern part of Inner Mongolia Autonomous Region. The Songhua River is the third-longest river in China and has three main tributaries, namely, Nenjiang River, Main Songhua River, and Second Songhua River (Khan et al., 2018; Zhao et al., 2018). The basins of the three tributary rivers include the Nenjiang Basin (NJ), the Downstream Songhua River Basin (SD), and the

Upstream Songhua River Basin (SU) (Figure 1). The Nenjiang River is 1370 km in length, and the corresponding drainage has an area of $2.55 \times 10^6$ thousand $km^2$. The Main Songhua River has a length of 939 km and the downstream catchment of the Songhua River Basin (SD) covers an area of $1.86 \times 10^6$ $km^2$. The Second Songhua River has a length of 958 km and the upstream catchment of the Songhua River Basin (SU) has an area of $6.19 \times 10^5$ $km^2$ (Chen et al., 2019; Yang et al., 2018). Temperate and cold temperate climates characterize the whole Songhua River Basin: winter is long and cold and spring is windy and dry. The annual average air temperature ranges between 3 to 5℃, while yearly precipitation ranges from 400 to 800 mm from the southeast to the northwest region (Wang et al., 2018; Wang et al., 2015).

[Figure 1 is added here]

## 2.2 Data Source

### 2.2.1 Ice phenology

The ice records were obtained from the annual hydrological report, including ice phenology, yearly maximum ice thickness of the river centre and the corresponding DOY. (Hydrographic bureau of Chinese Ministry of Water Resources, 2010-2015). There existed 50, 35 and 71 hydrological stations in the NJ, SU and SD basins, totalling 156 stations. Five lake ice phenology were available, and the definitions are listed as below (Duguay et al., 2015; Hydrographic bureau of Chinese Ministry of Water Resources, 2015) :

- Freeze-up start is considered the first day when the floating ice can be observed with temperatures below 0 ℃;
- Freeze-up end is the day when a steady ice carapace can be observed on the river, and the area of ice cover takes up more than 80% in the view range;
- Break-up start is the first day when ice melting can be observed with surface ponding;
- Break-up end is the day when the surface is mainly covered by open water, and the area of open water exceeds 20%;
- Complete frozen duration regards the ice cover duration when the lake is completely frozen during the winter, from freeze-up end to break-up start.

### 2.2.2 Ice thickness

We used ice thickness, snow depth, and air temperature from 120 stations for the period ranging from 2010 to 2015, to study changes in ice thickness and establish the regression model described below. 37, 28, and 55 stations were located in the NJ, SU and SD basins, respectively. The hydrological report also provided ice thickness, snow depth on ice, and air temperature on bank every five days from November through April, totalling 37 measurements in one cold season. The average snow depth ware derived from the mean of three or four measurements around the ice hole for ice thickness measurement without human disturbance (Hydrographic bureau of Chinese Ministry of Water Resources, 2015). To enhance the performance of the regression model, cumulative air temperature of freezing was derived from air temperature from November to March.

### 2.3 Data analysis

Our overall method can be summarized in the following steps: First, we used Kriging to spatially interpolate in situ measurements of ice phenology. Second, we used Moran's I spatial autocorrelation to identify spatial clusters based on the interpolated ice phenology data. Finally, we analysed the drivers of spatial and temporal variability of the river ice thickness for each cluster. To do so, we used the Bayesian linear regression to quantify the links between the river ice thickness and snow depth and air temperature.

### 2.3.1 Kriging

Kriging has been widely applied to spatially interpolate in situ measurements of ice phenology (Choinski et al., 2015; Jenson et al., 2007), such as freeze-up start, freeze-up end, break-up start, break-up end and complete frozen duration. The average values of five ice phenology were calculated during the periods from 2010 to 2015 and explored accordingly with the Geostatistical wizard of ArcGIS software. The interpolation results exhibited their spatial distribution. We chose the ordinary Kriging method and set variation function as the spherical model. Moreover, isophanes connecting locations with the same ice phenology were also graphed with the interpolation results (Paramasivam and Venkatramanan, 2019).

### 2.3.2 Moran's I

Moran's I aims to observe the spatial autocorrelation developed by Patrick Alfred Pierce Moran, and the spatial autocorrelation is characterized by a correlation in a signal among nearby locations in space (Li et al., 2020). We calculated the global and Anselin Local Moran's I of completely frozen duration and ice thickness in ArcGIS software environment. The Moran's I indicate whether the distribution of regional values is aggregated, discrete or random (Mitchell, 2005). A positive Moran's I indicate a tendency toward clustering while a negative Moran's I indicate a tendency of dispersion (Castro and Singer, 2006). The Anselin Local Moran's I statistic identified the clustered spots, and the statistically significant were evaluated by the combined thresholds of the z-score or p-values.

### 2.3.3 Bayesian linear regression

Ice thickness had been modelled by the air temperature and the snow depth using Bayesian linear regression, which has been widely adopted in hydrological and environmental analysis (Gao et al., 2014; Zhao et al., 2013). Bayesian linear regression views regression coefficients and the disturbance variance as random variables, rather than fixed and unknown quantities. This assumption leads to a more flexible model and intuitive inferences (Barber, 2008). The Bayesian linear regression model was implemented in two models: a prior probability model considered the probability distribution of the regression coefficients and the disturbance; a posterior model predicted the response using the prior probability mentioned below. Using k-fold cross validation, we divided the input dataset into 5 equal subsets or folds, and used 4 subsets as the training set and the remaining as the testing set. The performance of the regression model was evaluated with the determination coefficient ($R^2$) and the root mean square error (RMSE).

In this paper, we treated ice thickness on the river bank as the Y data, and snow depth over ice and air temperature as the X data with dataset size of 31. The ice thickness was measured on the riverbank every five days from November to March when the river was completely covered with ice with air temperature below 0℃. Air temperature and cumulative air temperature of freezing were considered in modelling. Additionally, the Pearson correlation was calculated to analyse the relationship between the five ice phenology events and ice-related parameters, including maximum ice thickness, snow depth on ice, and air temperature on the bank.

## 3 Results and discussion

### 3.1 Spatial variations of river ice phenology

The river ice phenology was analysed herein, including freeze-up start, freeze-up end, break-up start, break-up end, and complete frozen duration. The hydrological report only supplied one record of river ice phenology each year for all the 156 stations. For each hydrological station, the average values of five river ice phenology were calculated from the ice records from 2010 to 2015 and interpolated by the Kriging method to analyse the spatial distribution of the river ice phenology.

#### 3.1.1 Freeze-up end and break-up process

Figure 2 illustrates the average spatial distribution of the freeze-up start, the freeze-up end, and the isophanes in the Songhua River Basin of Northeast China from 2010 to 2015. Figure 3 shows the spatial distribution of the break-up start and the break-up end. The corresponding statistics are listed in Table 1. Freeze-up start ranged from October 28th to November 21st with a mean value of November 7th, and freeze-up end ranged from November 7th to December 8th with a mean value of November 22nd. Break-up start ranged from March 24th to April 20th with a mean value of April 9th, and break-up end ranged from March 31th to April 27th with a mean value of April 15th. These four parameters showed a latitudinal gradient: freeze-up start and freeze-up end decreased while break-up start and break-up end increased with the increase of latitude, except in the NJ basin. The middle part of the NJ basin had the highest freeze-up start and freeze-up end and decreased to the southern and northern parts. As the latitude decreased, the air temperature tended to increase, leading to later freeze-up and earlier break-up with shorter ice-covered duration, and vice versa.

[Figure 2 is added here]

[Figure 3 is added here]

[Table 1 is added here]

#### 3.1.2 Complete frozen duration

Figure 4(a) illustrates the average spatial distribution of complete frozen duration interpolated by kriging and the isophanes in the Songhua River Basin from 2010 to 2015. The complete frozen duration ranged from 110.74 to 163.00 days with a mean value of 137.86 days, increasing with latitude. Interestingly, the isophanes of complete frozen

duration had different directionality, increasing from the southeast to northwest, which could also be found in the other parameters. Both freeze-up start and freeze-up end correlated negatively with the latitude, with coefficients of -0.66 and -0.53, respectively (n=156, p < 0.001). However, the break-up start, the break-up end, and the complete frozen duration were all positively correlated with latitude with coefficients of 0.48, 0.57, and 0.55, respectively (n=156, p < 0.001). We built the linear regression equations between the river ice phenology and latitude. As the latitude increased by 1°, freeze-up start and freeze-up end occurred 2.56 and 2.32 day early, the break-up start and break-up end arrived 2.36 and 2.37 day late, causing an increase of 4.48 days for the complete frozen duration. This could be explained by the decreasing solar radiation with latitude influencing the ice thaw and melting processes directly.

The Global Moran's I statistic of the complete frozen duration was 1.36 with z scores and p value of 2.41 and 0.02, which indicated that complete frozen duration showed a clustered pattern with confidence level of 95% for the whole basin. Then Anselin local Moran's I was calculated to identify statistically significant spatial outliers for each hydrological location in Figure 4(c). Results showed that 14 of 156 hydrological stations showed a statistically significant cluster of high values, 17 of 156 showed a statistically significant cluster of low values and 124 of 156 showed no significant cluster at the 95 percent confidence level. Both global and local Moran's I indicated the high values of complete frozen duration clustered along with the Xiao Hinggan Mountains, and the low values of complete frozen duration grouped around the Changbai Mountains.

[Figure 4 is added here]

## 3.2 Variations of ice thickness

We explored the spatial pattern of ice thickness using the yearly maximum ice thickness gathered from 156 stations and examined the seasonal changes of ice thickness, snow depth on ice and air temperature based on the time series from November to April.

### 3.2.1 Spatial patterns of ice thickness

Figure 5 illustrates the spatial distribution of the yearly maximum ice thickness of the river centre and the corresponding DOY. Table 2 summarized the statistical result of maximum ice thickness and the DOY. Maximum ice thickness ranged from 12 cm to 146 meter, with an average value of 78 cm. The maximum ice thickness between 76 and 100 cm accounted

for the most significant percentage of 43.33%, followed by 31.67% of maximum ice thickness between 50 and 75 cm. As shown in Table 2, five stations had a more exceptional maximum ice thickness than 125 cm. The DOY of maximum ice thickness had an average value of February 21st, and maximum ice thickness mainly occurred 59 and 40 times in February and March, respectively. Four of the five highest maximum ice thickness greater than 125 cm happened in March, which is consistent with the inter-annual changes in ice development shown in Figure 6. The results suggested that the river ice was always the thickest and the steadiest in February or March, which has important implications for human activities, such as ice fishing and entertainment. The ice thickness didn't show the same latitudinal distribution as ice phenology, which suggested that more climate factors should be taken into consideration, such as snow depth and wind speed.

[Figure 5 is added here]

[Table 2 is added here]

### 3.2.2 Seasonal changes of ice thickness

Figure 6 displays the seasonal changes of ice development using ice thickness, average snow depth on ice, and air temperature, which was collected on bank every five days from November to April during the period between 2010 and 2015. The variations of ice characteristics differed significantly due to time and location.   Among the three basins, the NJ basin had the highest snow depth of -29.15 ± 9.99℃, followed by -25.61 ± 9.02 ℃ of the SD basin, and -22.17 ± 7.33 cm of the SU basin. The SD basin had the highest snow depth of 9.18 cm ± 3.39 cm on the average level, followed by 8.35 cm ± 4.60 cm of the SU basin, and 8.23 cm ± 3.92 cm of the NJ basin. The changes in daily ice thickness and snow depth had a similar overall trend, while air temperature followed the opposite pattern. Both ice thickness and snow depth increased from November and reached a peak in March, then dropped at the beginning of April. The air temperature showed a distinct trend and reached the bottom in the middle of February, which is earlier than the peaks of maximum ice thickness and snow depth. In Figure 6, the day when ice thickness reached the maximum value was 50, 54 and 60 days later than the day when air temperature reached the lowest value in the NJ, SU and SD basin respectively.

[Figure 6 is added here]

### 3.3 The relationship between ice regime and climate factors

### 3.3.1 Correlation analysis

Figure 7 displays the correlation matrix between lake ice phenology events and three ground measurements from 120 hydrological stations. The lake ice phenology events included the freeze-up start, the freeze-up end, the break-up start, the break-up end, and the complete frozen duration. The three ground measurements covered the yearly mean values of snow depth, the air temperature on bank, and the maximum ice thickness. The colour

intensity and sizes of the ellipses are proportional to the correlation coefficients. The maximum ice thickness had a higher correlation with four of the five indices than snow depth and air temperature on the bank, except with freeze-up start. The maximum ice thickness and break-up end had the highest correlation of 0.63 (p<0.01, n=120). During the freeze-up process, two freeze-up dates had a negative association with the maximum ice

thickness and snow depth. During the break-up process, two break-up dates had positive correlations with maximum ice thickness and snow depth. The complete frozen duration showed a positive correlation with the maximum ice thickness and the snow depth. The situation of air temperature was contrary to that of the maximum ice thickness and air temperature. Regarding the annual changes, no significant correlation was found between

snow depth and five ice phenology events in Figure 7.

[Figure 7 is added here]

Figure 8 shows the bivariate scatter plots between the yearly maximum ice thickness and the ice phenology along with regression equations attached. The break-up process had a negative correlation with the maximum ice thickness, while the freeze-up had a positive

correlation. Besides, the break-up process had a higher correlation with the maximum ice thickness, and the break-up end had the highest correlation coefficients with the maximum ice thickness of 0.65 (p<0.01). The complete frozen duration also had a positive correlation with maximum ice thickness of 0.57 (P<0.01), which means that a thicker ice cover in winter can lead to a delay for the melting time in spring. The break-up depends on not only

the spring climate conditions but also influenced by the ice thickness during last winter. A thicker ice cover stores more heat in winter, taking a longer time to melt in spring (Yang et al., 2019). The limited performance of the regression model can be attributed to the difficulties in determining river ice phenology. Although a uniform specification for ice regime observations was required, the inhomogeneities among different stations could not

be ignored.

[Figure 8 is added here]

To further explore the role of snow cover, the monthly correlation coefficients between the ice thickness, the snow depth and the air temperature on bank were calculated and listed in Table 3. The correlation coefficients between the ice thickness and the snow depth increased from November to March and reached a peak of 0.75 in March when ice was the thickest. This indicated an increasingly important role of the snow depth on the ice thickness as the ice accumulated. The higher correlation coefficients between the ice thickness and the air temperature on bank in November and December revealed that the air temperature played a more critical role in the freeze-up process. The positive correlation coefficient between snow depth and ice thickness (Table 3) showed two opposite effects of the snow depth during the ice development. During the ice-growth process, snow depth protects the ice from cold air and slows down the growth rate of ice thickness. During the ice-decay process, the lake bottom ice stops to grow, and the surface snow or ice melts, and slush forms accordingly. The melting speed depends on the ability to absorb heat, and the slush can absorb more heat, which would promote melting (Kirillin et al., 2012). The slush often existed in multiple freeze-thaw cycles of river ice before it completely disappears. Therefore, when studying the role of snow cover, the status of river ice could not be neglected.

[Table 3 is added here]

### 3.3.2 Regression modelling

We carried out cross-validation for Bayesian linear regression using k-fold method and set K value as 5. For each iteration, a different fold was held out for testing, and the remaining 4 subsets were applied for training. The training and testing were repeated for five iterations. Table 4 lists the $R^2$ of the training and testing process for each iteration. The best Bayesian linear regression was determined when the bias between testing and training regression was the smallest, and the corresponding $R^2$ were marked as bold and red, as shown in Table 4.

Figure 9 illustrates the scatter plot between the measured and the predicted ice thickness with Bayesian linear regression in three sub-basins in Northeast China. From Figure 9, the $R^2$ of Bayesian linear regression varies from 0.81 to 0.95, and RMSE varies from 0.08 to 0.18 meters. The model works best in the SU basin, followed by the NJ and the SD basins. Figure 9 indicates that the snow depth outweighs the air temperature in terms of the effect on ice thickness, which is consistent with previous studies (Magnuson et al., 2000; Sharma

et al., 2019). Moreover, replacing air temperature on bank with cumulative air temperature
of freezing enhanced the model performance in all three basins, revealing a more important
role of cumulative air temperature of freezing than air temperature. For the Bayesian linear
regressions, we used the field measurements that spanned from November to March, thus
focusing only on the cold part of the year. During this period, the river surface is completely
frozen, and the air temperature that falls below 0℃ promotes the ice growth. April is the
month when the rise of air temperatures above 0℃ enables the river ice to melt.

[Figure 9 is added here]

The correlation between air temperature and ice regime in Figure 7 was not as significant
as found in some previous studies (Park et al., 2016; Stefan and Fang, 1997). One of the
reasons is that previous studies often averaged the air temperatures over a longer period
and at a regional scale, therefore losing the signal on seasonality at a local scale (Pavelsky
and Smith, 2004; Yang et al., 2020). To circumvent this shortcoming, we applied the
regression analysis on seasonal time series of ice thickness and air temperature. Our work
considered this and established the regression using the seasonal time series of ice thickness
and air temperature. When building the Bayesian regression equation, the increasing $R^2$
displayed that the cumulative air temperature of freezing behaved better than the air
temperature on bank, which suggested that heat exchanges between river surface and
atmosphere dominated the ice process. Heat loss is mainly made up of sensible and latent
heat exchange (Beltaos and Prowse, 2009; Robertson et al., 1992) , which is proportional
to the cumulative air temperature of freezing  during the cooling process. During the
complete frozen duration, the snow depth, along with the wind speed began to influence
the heat exchange and ice thickening. Air temperature exerted a lesser vital effect on spring
break-up, which is more dependent on the ice thickness and the snow depth. In summary,
snow depth dominated the ice process when the river was completely frozen. At the same
time, the cumulative air temperature dominated during the transition process between open
water and completely frozen condition.

**4 Conclusions**

Five river ice phenology proxies, including freeze-up end, freeze-up start, break-up end,
break-up start, and complete frozen duration in the Songhua River Basin of Northeast China,
have been investigated using in situ measurements for the periods from 2010 to 2015.

According to the spatial distribution interpolated by the ordinary Kriging method, the river ice phenology indicators followed the latitudinal gradient and a changing direction from southeast to northwest. As the latitude increased by 1°, the freeze-up start and can freeze-up end happened 2.56 and 2.32 day earlier, the break-up start and break-up end arrived 2.36 and 2.37 days later, resulting in 4.48 days increase for complete frozen duration.

The spatial autocorrelation of the completely frozen duration and maximum ice thickness has been explored by global and Anselin Local Moran's I. The Global Moran's I with a z score of 1.36 showed that the complete frozen duration showed a clustered pattern at the 95% confidence level. In contrast, the maximum ice thickness didn't show a significantly clustered pattern. The Anselin local Moran's I result indicated that the high values of complete frozen duration clustered along the Xiao Hinggan Mountains, and the low values of the complete frozen clustered in the Changbai Mountains. The maximum ice thickness over 125 cm was distributed along with the ridge of Da Hinggan Mountains and Changbai Mountains, and maximum ice thickness occurred most often in February and March during the cold season.

Based on the analysis of monthly time series measurements, snow cover played an increasingly important role as the river becomes completely frozen. The temporal variability in air temperature was more correlated with the variability in ice phenology while snow depth was more correlated with ice thickness. Six Bayesian regression models were built among the ice thickness and the air temperature and the snow depth in three sub-basins of the Songhua River, considering air temperature, as well as cumulative air temperature. Results showed that snow cover correlated with ice thickness significantly and positively during the periods when the freshwater was completely frozen. In line with the performance metrics ($R^2$, root mean square error), the cumulative air temperature of freezing was shown to provide a better predictor than the air temperature in simulating the ice thickness changes compared with the air temperature.

This study provides a quantitative investigation of the ice regime in the Songhua River Basin of Northeast China and established potential regression models for projecting future changes in the ice regime. Remote sensing data could provide long-term and wide-range information for ice thickness and ice phenology since 1980. Data analysed in this study presents a valuable reference for future studies that rely on remote sensing observations of

the river ice thickness in this area. Therefore, we plan to use satellite data to enlarge our study scope in our future work.

## Author Contribution

Song K.S. and Yang Q. designed the idea of this study together. Yang Q. and Wen Z.D. wrote the paper and analysed the data cooperatively; Hao X.H. provided valuable

suggestions for the structure of study and paper; Li W.B. and Tan Y. exerted efforts on data processing and graphing. This article is a result of collaboration with all listed co-authors.

## Competing interest

The authors reported no potential conflict of interest.

## Acknowledgments

The research was sponsored by the National Natural Science Foundation of China (41801283, 41971325, 41730104). The anonymous reviewers to improve the quality of this manuscript are much appreciated.

## Data availability

The data that support the findings of this study are available from the corresponding author

upon reasonable request.

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

**Tables**

Table 1 Summary statistics of ice phenology interpolated by Kriging from 2010 to 2015. The ice phenology indicators included freeze-up start (FUS), freeze-up end, break-up start (BUS), break-up end (BUE), complete frozen duration (CFD). NJ, SD and SU represent the Nenjiang Basin, the downstream Songhua River Basin (SD) and the upstream Songhua River Basin (SU). DOY denotes day of year. Std Dev. denotes standard deviation.

| Basins | Statistics | FUS (DOY) | FUE (DOY) | BUS (DOY) | BUE (DOY) | CFD (day) |
|--------|-----------|-----------|-----------|-----------|-----------|-----------|
| NJ | Maximum | 319.14 | 334.98 | 110.54 | 117.61 | 163.00 |
|    | Mean | 307.02 | 324.58 | 98.65 | 106.64 | 139.39 |
|    | Minimum | 301.41 | 311.30 | 84.53 | 90.40 | 119.11 |
|    | Std Dev. | 3.91 | 5.69 | 8.16 | 6.80 | 13.22 |
| SD | Maximum | 321.08 | 334.36 | 110.01 | 102.84 | 154.06 |
|    | Mean | 313.74 | 326.70 | 102.55 | 97.15 | 140.86 |
|    | Minimum | 305.64 | 316.80 | 93.22 | 92.37 | 125.32 |
|    | Std Dev. | 2.83 | 3.13 | 3.92 | 2.12 | 5.69 |
| SU | Maximum | 325.92 | 342.09 | 98.25 | 114.37 | 133.62 |
|    | Mean | 320.39 | 334.35 | 91.93 | 106.43 | 122.61 |
|    | Minimum | 313.79 | 327.68 | 83.46 | 95.69 | 110.74 |
|    | Std Dev. | 2.34 | 3.09 | 3.21 | 4.24 | 4.85 |
| Total | Maximum | 325.92 | 342.09 | 110.54 | 117.61 | 163.00 |
|    | Mean | 311.16 | 326.58 | 99.25 | 105.38 | 137.86 |
|    | Minimum | 301.41 | 311.30 | 83.46 | 90.40 | 110.74 |
|    | Std Dev. | 5.74 | 5.54 | 7.17 | 6.34 | 11.68 |


Table 2 The frequency of yearly maximum ice thickness from November to April. The column represents different months in cold season and the row represents yearly maximum ice thickness with the unit of centimeter.

| Month \ MIT | <50 | 51-75 | 76-100 | 101-125 | 126-150 |
|---|---|---|---|---|---|
| December | 4 | 1 | 0 | 1 | 0 |
| January | 4 | 4 | 1 | 0 | 0 |
| February | 4 | 25 | 26 | 3 | 1 |
| March | 1 | 3 | 24 | 8 | 4 |
| April | 0 | 2 | 1 | 0 | 0 |
| After April | 0 | 3 | 0 | 0 | 0 |
| Total | 13 | 38 | 52 | 12 | 5 |


Table 3 Correlation coefficient between maximum ice thickness (MIT) and average snow depth (ASD), and air temperature on bank (BAT) with a dataset size of 120 stations. The asterisk indicates the significant level of correlation coefficients, ** means significant at 99% level ($p<0.01$), and * means significant at 95% level ($p<0.05$).

| Correlation Coefficients | November | December | January | February | March |
|---|---|---|---|---|---|
| MIT vs. ASD | 0.17 | 0.66* | 0.53* | 0.59* | 0.75** |
| MIT vs. BAT | -0.90** | -0.80** | -0.55* | -0.30 | -0.45 |


Table 4 The cross validation of Bayesian linear regression using k-fold method and the K value was set as 5. The $R^2$ values of training dataset and testing dataset based on the Bayesian regression. Ice thickness was treated as dependent variables, and air temperature, snow depth on ice as independent variables. Air temperature and cumulative air temperature of freezing were considered in the model building.

| Basin | Air temperature | | Cumulative air temperature | |
|---|---|---|---|---|
| | Training | Testing | Training | Testing |
| NJ | 0.80 | 0.99 | 0.84 | 0.99 |
| | 0.89 | 0.80 | 0.90 | 0.86 |
| | 0.84 | 0.92 | 0.89 | 0.82 |
| | 0.90 | 0.56 | 0.91 | 0.61 |
| | **0.85** | **0.91** | **0.89** | **0.89** |
| SU | 0.83 | 0.92 | 0.95 | 0.98 |
| | 0.83 | 0.65 | 0.96 | 0.83 |
| | 0.81 | 0.94 | 0.95 | 0.99 |
| | 0.84 | 0.79 | **0.95** | **0.93** |
| | **0.82** | **0.82** | 0.94 | 0.98 |
| SD | 0.80 | 0.96 | 0.82 | 0.98 |
| | 0.84 | 0.16 | 0.86 | 0.25 |
| | 0.81 | 0.84 | 0.82 | 0.87 |
| | 0.79 | 0.97 | 0.79 | 0.96 |
| | **0.81** | **0.80** | **0.82** | **0.83** |

**Figures**

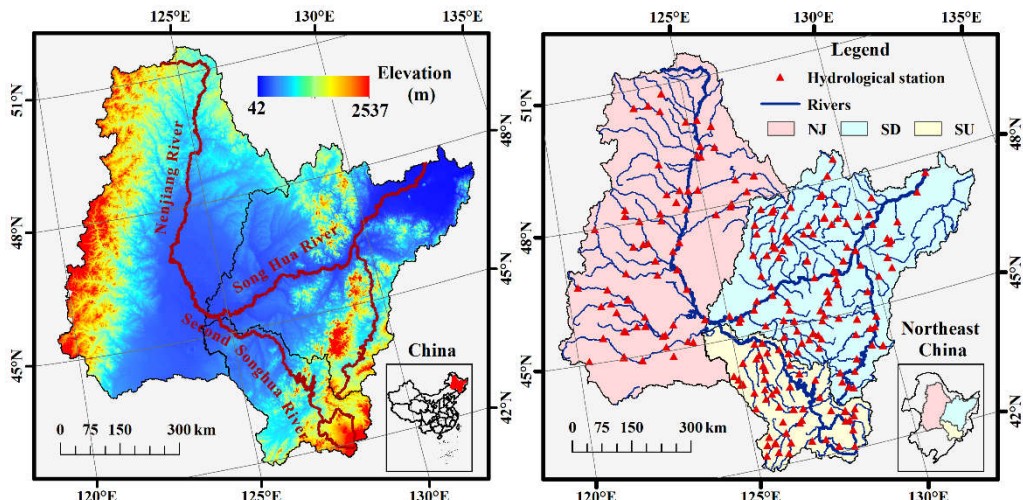

Figure 1 The geographic location of the Songhua River Basin showing (a) the elevation and (b) the location of 156 hydrological stations. The Songhua River Basin includes three sub-basins: Nenjiang River Basin (NJ), downstream Songhua River Basin (SD) and upstream Songhua River Basin (SU). Elevation data are from the Shuttle Radar Topography Mission (SRTM) with spatial resolution of 90 meters.


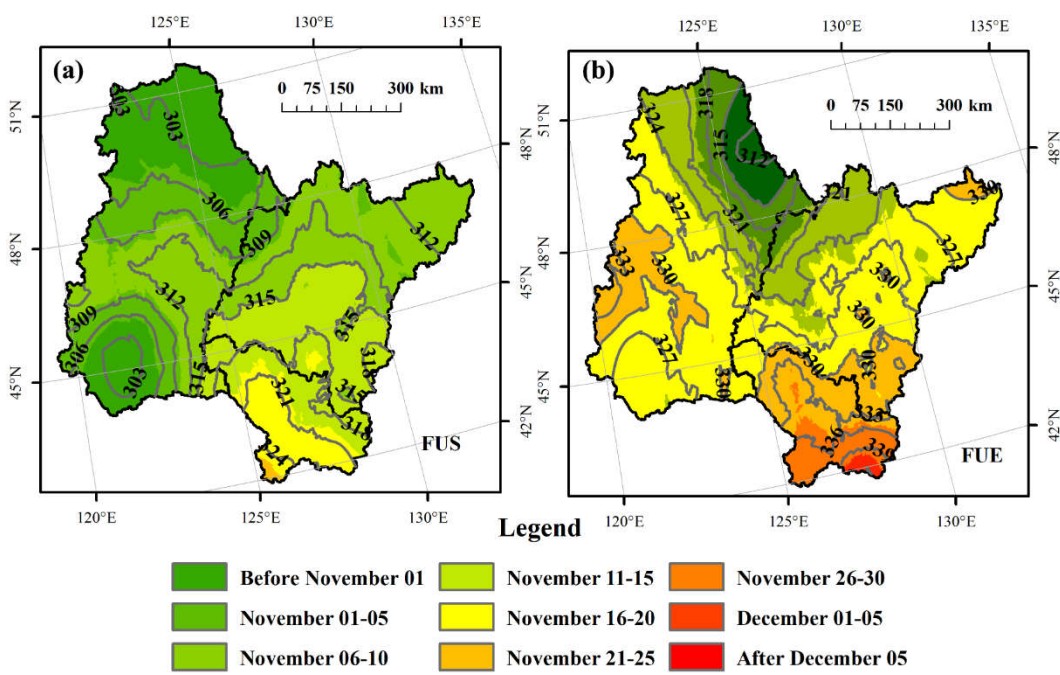

Figure 2 The average spatial distribution of freeze-up start (FUS) (a) and freeze-up end (FUE) (b) in the Songhua River Basin of Northeast China from 2010 to 2015. The number labels indicate the day of year (DOY) of the isophenes.


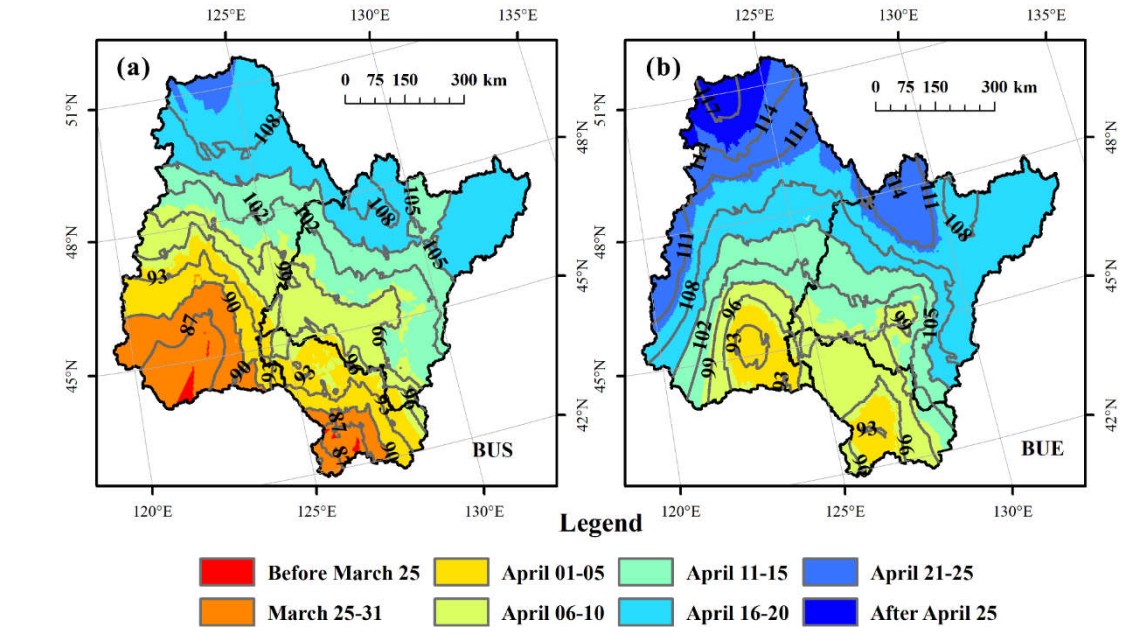

Figure 3 The average spatial distribution of break-up start (BUS) (a) and break-up end (BUE) (b) in the Songhua River Basin of Northeast China from 2010 to 2015. The number labels indicate the day of year (DOY) of the isophenes.

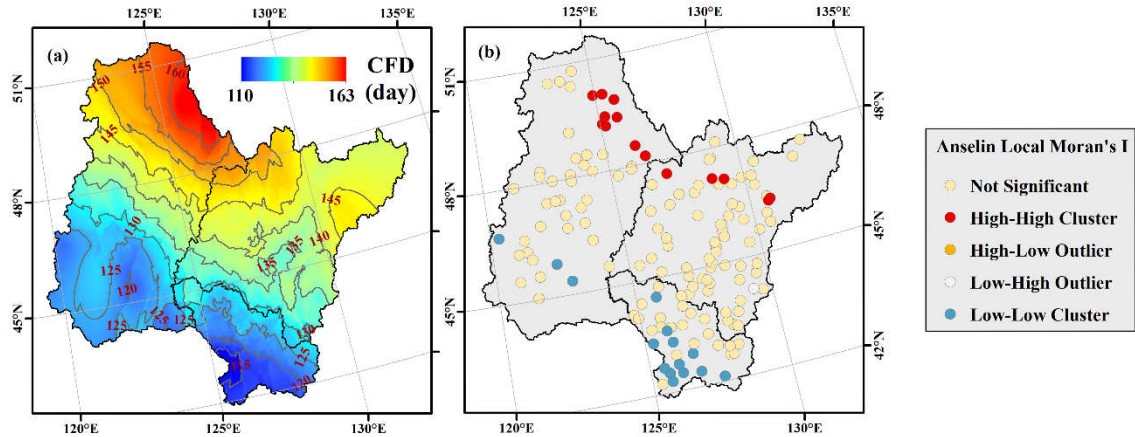

Figure 4 The spatial distribution of complete frozen duration (a) interpolated using Kriging method and Anselin local Moran's I (b) in the Songhua River Basin of Northeast China.

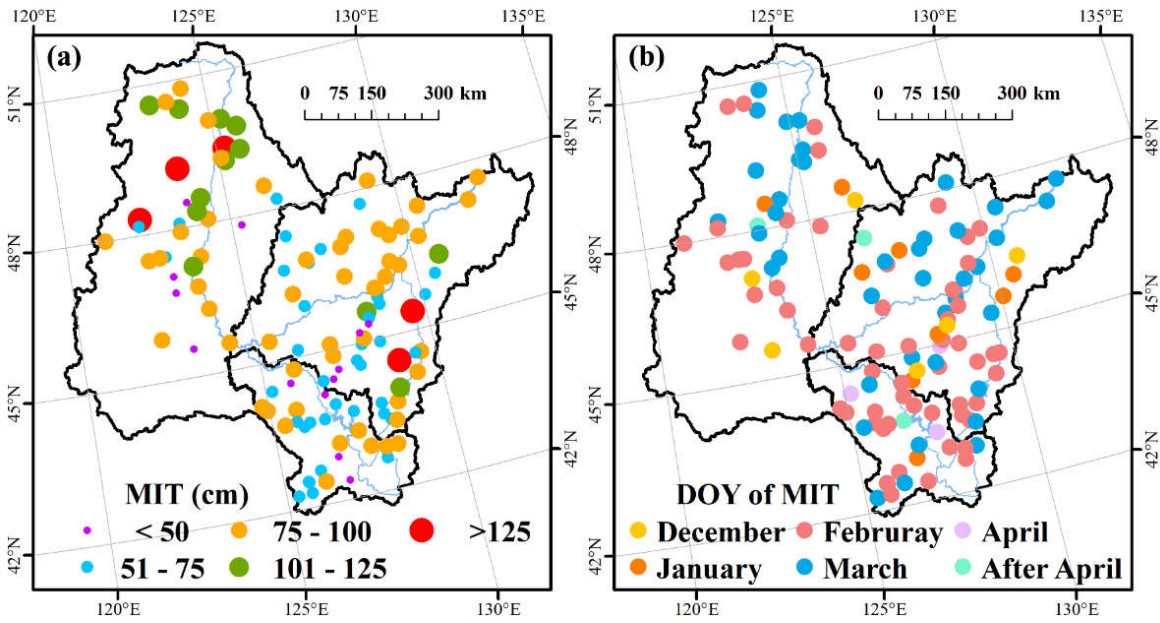

Figure 5 The spatial distribution of yearly maximum ice thickness (MIT) of the river centre (a) and the corresponding date (b).

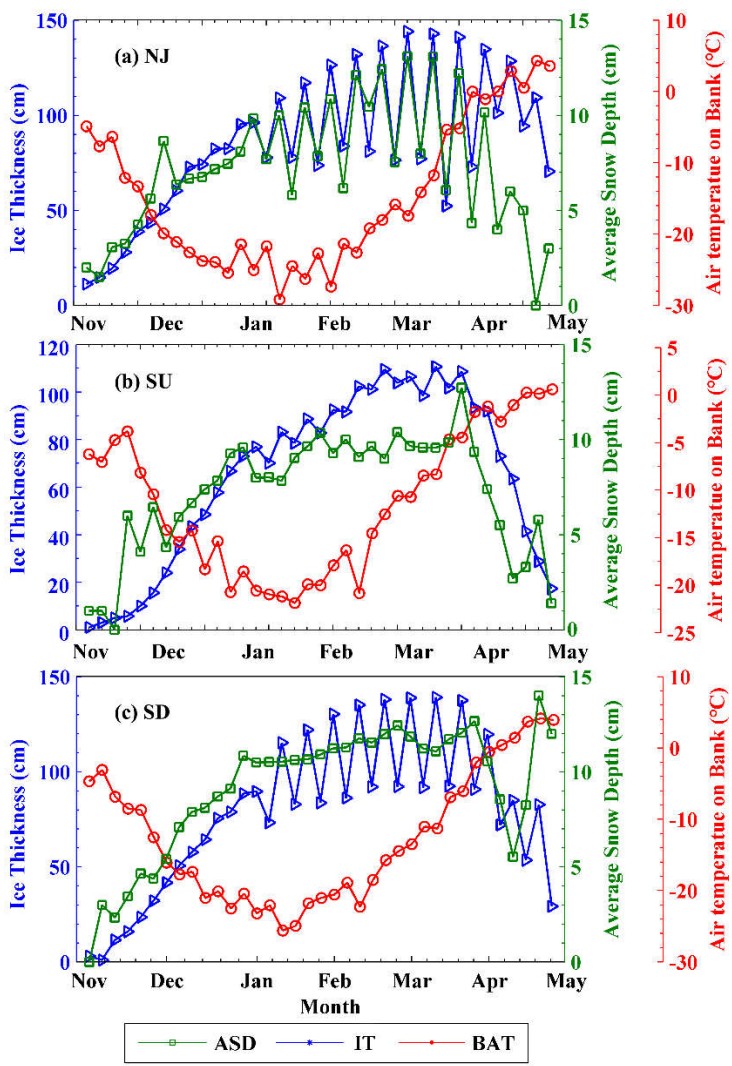

Figure 6 Average seasonal changes in ice thickness (IT), average snow depth (ASD) and air temperature on bank (BAT) from November to April for the period 2010 - 2015.

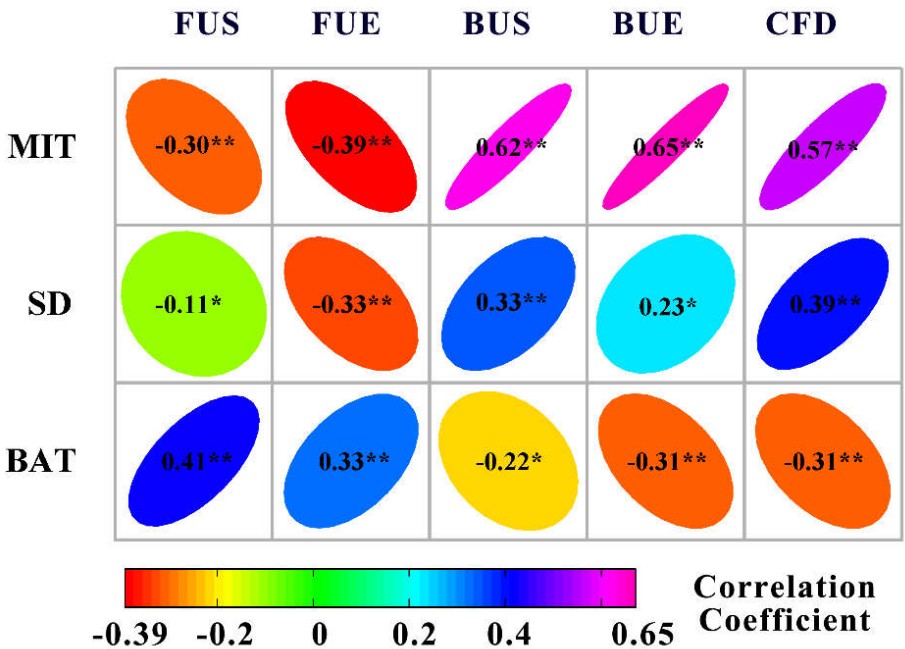

Figure 7 Correlation matrix between maximum ice thickness (MIT), snow depth (SD) and air temperature on bank (BAT) and lake ice phenology events with data from 120 stations. The asterisk indicates the significance level of the correlation coefficients, ** means significant at 99% level ($p<0.01$), and * means significant at 95% level ($p<0.05$).

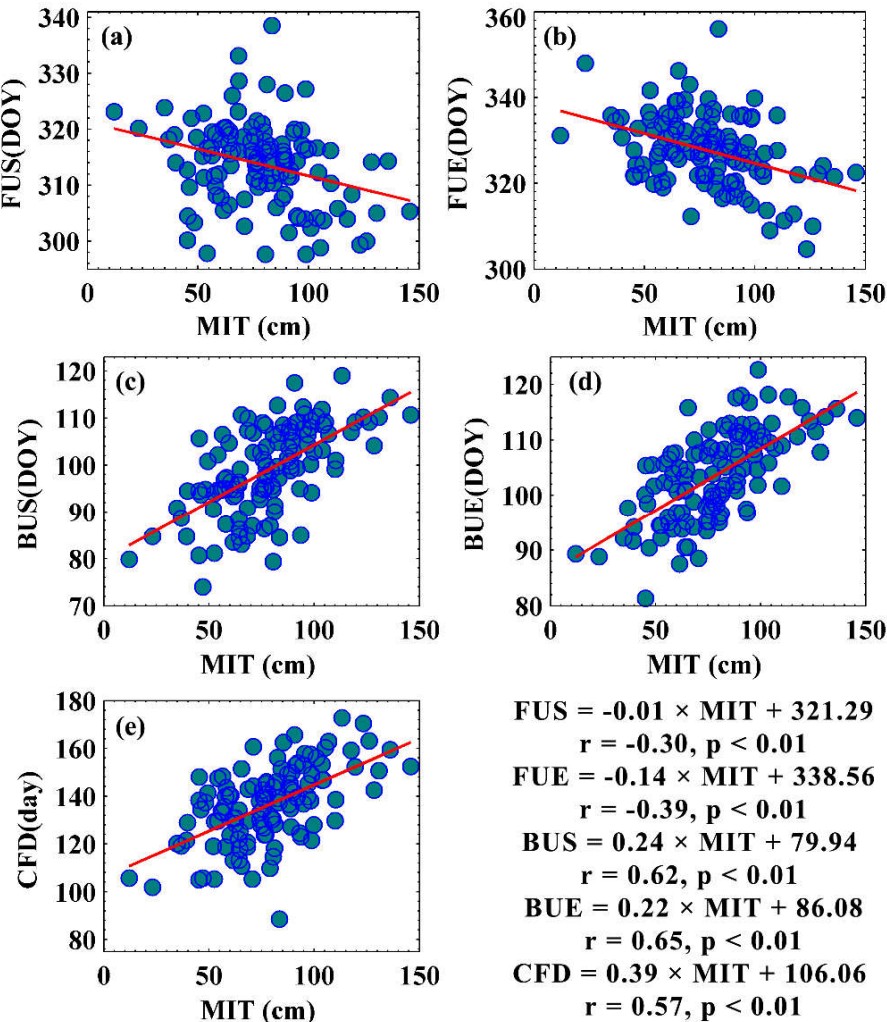

Figure 8 The bivariate scatter plots with linear regression lines between yearly maximum ice thickness (MIT) and ice phenology with dataset size of 120; r and p denote the correlation coefficient and p value of the regression line. The ice phenology events include freeze-up start (FUS), freeze-up end (FUE), break-up start (BUS), break-up end (BUE) and complete frozen duration (CFD).

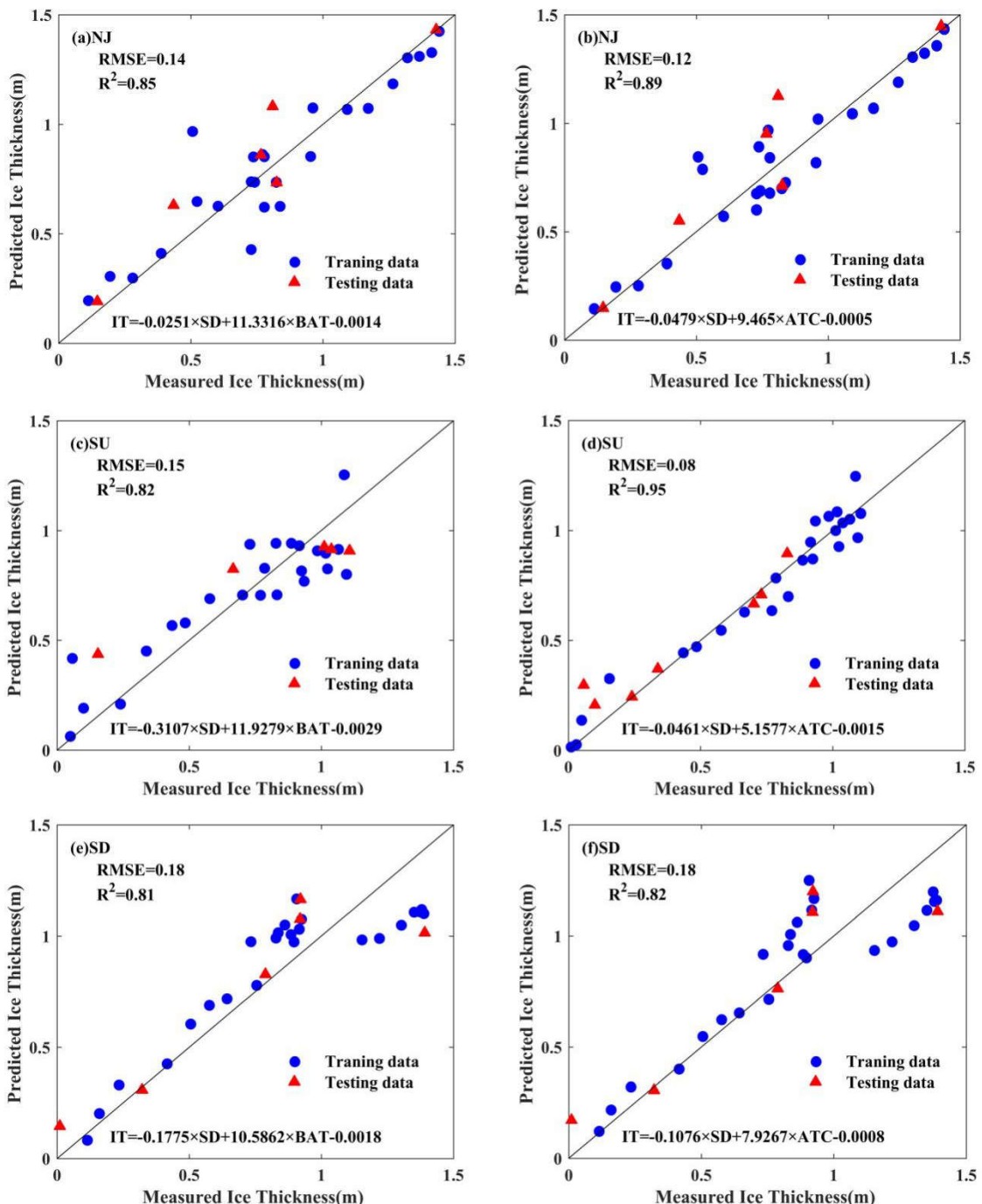


Figure 9 Scatter plots between measured and predicted ice thickness using Bayesian linear regression in three sub-basins (NJ: Nenjiang Basin, SU: upstream Songhua River Basin, and SD: downstream Songhua River Basin) in Northeast China. The model treated ice thickness as the independent variable, and snow depth and air temperature as dependent

variables. Two types of air temperature were used: BAT represents air temperature on bank; ATC represents cumulative air temperature of freezing.