# Peer review of "Investigation of spatial and temporal variability of river-ice phenology and thickness across Songhua River Basin, Northeast China"

_The Cryosphere, 2019_

## Referee Comment (RC1) · Anonymous Referee #1 · 9 Jan 2020

The paper mainly investigates the relationship between ice thickness and ice phenology in three sub-basins of Songhua River in the time interval from 2010 to 2015. The study area is selected as it is a sensitive area to global warming in Northeast China. Application for the obtained results could be in validating the retrieval of ice development by remote sensing.

As general comment the paper present many data, but their discussion is not exhaustive and critical. The introduction is not too effective in presenting the study and its rationale. I therefore suggest to significantly improve Sec 1 and Sec 4.

Other comments: - Sec 2.2: In situ data analyzed are not fully descripted. Their temporal resolution is 5 days, but what about the spatial one? - line 109-11: FUE and BUE have a standard definition? In this case reference is missing, otherwise justify the

20%. - Sec 2.3: Some comment about the two methodologies adopted are needed, i.e. applicability limits, reliability of results, pros, cons. . . - Figures and caption are generally not consistent. As an example figure 2 and 3 use both notation DOY and dates, and the label (a) and (b) are never cited in caption. - line 179: PC not yet defined - Lines 197-200: Figure 5 does not show DOY - some English revision is needed in the manuscript, for example lines 23-25,40-43, 215,299. . .

---

## Referee Comment (RC2) · Anonymous Referee #2 · 9 Jan 2020

The paper presents the results of a regression analysis between snow depth and air temperature; and ice phenology and thicknesses using data from 2010 to 2015 for the Songhua River basin in Northeast China. The results of the correlation showed high correlation between snow depth and ice thickness as well as high correlation between air temperature and ice phenology (freeze-up, break-up and mid-winter). These conclusions are basic scientific knowledge for freshwater ice scientists and have been well established over the years. The data can be used for more advanced analysis than just linear regression between basic parameters. The overall structure of the paper is not well laid out. There is no literature review describing what is the current state of knowledge and what is this manuscript adding to the research community. Also, the objectives and how the results of this exercise will be used are not presented. The anal-

ysis section is brief and does not explain the details of the data processing. It wasn't clear to me how the hydrometric data was gathered (resolution, methods, accuracy. . .) and processed and how the dates of ice phenology were extracted from satellite images. The conclusions are brief and is not inclusive.

On a separate note, the language of the manuscript overall is not clear and it seems the text was not read-proof before submission with many repeated sentences (line 55, 60, 70. . .), many grammatical and spelling errors.

In conclusion, the current version of the manuscript describes with historical data the already established basic science of ice thickening, growth/deterioration of the cover with some correlation between meteorological (snow depth and air temperature) conditions and the dates of freeze-up or break-up. I cannot see any scientific contribution to the ice research community. Therefore, I recommend rejection of this version of the manuscript.

---

## Author Comment (AC1) · 7 Mar 2020

Thank you for the helpful suggestions and we have carefully gone through the manuscript and improved accordingly. Our replies are listed as bellows.

The results of the correlation showed high correlation between snow depth and ice thickness as well as high correlation between air temperature and ice phenology (freeze-up, break-up and mid-winter). These conclusions are basic scientific knowledge for freshwater ice scientists and have been well established over the years.

Reply: It seems that the relationship between the ice regime and the impact facts are common sense and have been well established. Still, different scholars held a considerably different opinions on this issue. Generally, snow depth plays a more crucial

role than air temperature (Morris et al., 2005) and increasing snow depth provide a favorable condition for thicker ice cover. In comparison with other works, the air temperature had more effect on ice thickness than snow depth and attributed this to the high snowfall of study area (Gao and Stefan, 2004). Shiklomanov and Lammers (2014) documented that in situ observations at Russian river mouths where ice thickness decreased had not revealed any significant correlation between ice thickness and SND. Whether snow depth or air temperature is the primary factor influencing ice formation and decay deserves further exploration in Northeast China.

Thank you for this enlightened question. We have improved the Abstract and Conclusions to emphasize our findings in this paper. According to high correlation coefficients between maximum ice thickness and ice phenology, ice phenology closely related to ice thickness, primarily the break-up process. Compared with ice thickness, air temperature associated with ice phenology more closely. No significant correlation existed between snow depth and ice thickness unless the air temperature was falling below the freezing point. We conclude that snow depth is the primary factor when the river ice is completely frozen, and air temperature is the primary factor during the transition periods between ice-covered and open water.

Reference: Bian, Y., Yue, J., Gao, W., Li, Z., Lu, D., Xiang, Y., and Chen, J.: Analysis of the Spatiotemporal Changes of Ice Sheet Mass and Driving Factors in Greenland, Remote Sensing, 11, 10.3390/rs11070862, 2019.

Gao, S., and Stefan, H. G.: Potential Climate Change Effects on Ice Covers of Five Freshwater Lakes, Journal of Hydrologic Engineering, 9, 226-234, doi:10.1061/(ASCE)1084-0699(2004)9:3(226), 2004.

Morris, K., Jeffries, M., and Duguay, C.: Model simulation of the effects of climate variability and change on lake ice in central Alaska, USA, Annals of Glaciology, 40, 113-118, 2005

Shiklomanov, A. I., and Lammers, R. B.: River ice responses to a warming Arctic–recent evidence from Russian rivers, Environmental Research Letters, 9, 035008, https://doi.org/10.1088/1748-9326/9/3/035008, 2014.

The data can be used for more advanced analysis than just linear regression between basic parameters.

Reply: The data are further analyzed as you suggested.Firstly, we analyzed the ice phenology using rotated empirical orthogonal function and found five typical geographic zones. Secondly, we analyzed the spatiaol changes and inner annual changes of ice thickness. Thirdly, we supplement the correlation between ice thickness and snow depth, and air temperature in the view of three basins. Besides, we adopted accumulated air temperature in this study, and found that accumulated temperature had a higher correlation with ice thickness compared with air temperature.

The overall structure of the paper is not well laid out.

Reply: We adjusted the structure of this paper, and the new structure is listed as follows.

1 Introduction

2 Materials and methods

2.1 Study area

2.2 Data source

2.3 Data analysis

2.3.1 Kriging intepolation

2.3.2 Rotated empirical orthogonal function (REOF)

2.3.3 Partial least squares regression

3 Results

3.1 General distribution of ice phenology

3.2 Changes in ice thickness

4 Discussion

4.1 Error sources

4.2 The relationship between ice thickness and ice phenology

4.3 The role of snow and air temperature

5 Conclusions

There is no literature review describing what is the current state of knowledge and what is this manuscript adding to the research community.

Reply: Thanks for the concern, we have updated the introduction as you suggested based on literature review and supplements new references and emphasized on the diversity opinions on the role of snow cover during the ice process.

Also, the objectives and how the results of this exercise will be used are not presented.

Reply: We have modified the last paragraph of the introduction and suggested the possible application. The surface-based networks, including climatic and hydrological stations, have been established for tracing climate and hydrological changes in Northeast China, which are limited by the accessibility of surface-based networks and the range of filed measurement. To evaluate the influence of ice regime on regional climate and human environment, a robust investigation and quantitative analysis on ice process is necessary, which provide helpful information for projecting future changes in ice regime. The previous work explored the ice process in at one or more locations on a given river and ignored the changing regional pattern of ice development due sparse location. The objectives of this study are to: (1) examine and compare ice phenology dynamics of three sub-basins of Songhua River from 2010 to 2015; (2) explore the relationship between ice thickness and ice phenology; and (3) analyze the primary factor

influencing ice regime and ice thickness.

The analysis section is brief and does not explain the details of the data processing. It wasn't clear to me how the hydrometric data was gathered (resolution, methods, accuracy...) and processed and how the dates of ice phenology were extracted from satellite images.

Reply: Thank you for the comments and suggestions. This paper used field measurement rather than satellite data for further research. The in-situ lake ice records were available provided by the Chinese Ministry of Water Resources from 2010 to 2015, including ice phenology, ice thickness, snow depth on ice and air temperature on bank (BAT) (Annual hydrological report, 2010-2015).

Reference: Hydrographic bureau of Chinese Ministry of Water Resources, 2010-2015. Annual hydrological report: hydrological data of Heilongjiang River Basin. (in Chinese)

The conclusions are brief and is not inclusive. Reply: We have significantly improved the conclusion, as you suggested. Later on, we will upload the revised manuscript, you can check it out through the revised MS.

On a separate note, the language of the manuscript overall is not clear and it seems the text was not read-proof before submission with many repeated sentences (line 55, 60, 70...), many grammatical and spelling errors(Bian et al., 2019).

Reply: We have carefully revised the manuscript according to the reviewers' comments, and employed an English-language editing service Editsprings, to polish our wording.

─────────────────────────────

---

## Author Comment (AC2) · 7 Mar 2020

Thanks for those valuable and helpful comments. We have carefully revised the manuscript according to the reviewers' comments.

As general comment the paper present many data, but their discussion is not exhaustive and critical. The introduction is not too effective in presenting the study and its rationale. I therefore suggest to significantly improve Sec 1 and Sec 4.

Reply to comment: We have significantly improved Sec1 and Sec 4 as you suggested. Further, we did additional literature reviews, and provided the knowledge gap, and rationale for conducting the research.

Sec 2.2: In situ data analyzed are not fully descripted. Their temporal resolution is 5

days, but what about the spatial one?

Reply to comment: We introduced the station numbers. Besides, we added a new citation for the data source as follows, which similar to meteorological station, only represented several hundreds square kilometers.

Reference: Hydrographic Bureau of Ministry of Water Resources of the People's Republic of China, 2010-2015. Annual hydrological report: hydrological data of Heilongjiang River Basin. (in Chinese)

Line 109-11: FUE and BUE have a standard definition? In this case reference is missing, otherwise justify the 20%.

Reply to comment: A new citation on annual hierological report has been inserted, and explain the data source more clearly. The definitions of ice phenology refer to specification for observation of ice regime in rivers (2015) and are further defined in details: freeze-up start (FUS) is considered as the first day when floating ice can be observed with temperature below 0 ℃; freeze-up end (FUE) is considered as the day when steady ice cover across the river called ice carapace can be observed, and the area of ice cover is more than 80% of view range; break-up start (BUS) is considered as the first day when ice melting could be observed with surface ponding; break-up end (BUE) is considered as the day when the surface is mainly covered by open water and the area of open water exceed 20%; complete frozen duration (CFD) is the ice cover duration when the lake is completely frozen during the winter, staring from FUE to BUS.

Reference: Cai, Y., Ke, C.Q., Yao, G., and Shen, X.: MODIS-observed variations of lake ice phenology in Xinjiang, China, Climatic Change, 10.1007/s10584-019-02623-2, 2019.

Specification for observation of ice regime in rivers (SL59-2015), Ministry of Water Resources of the People's Republic of ChinaïijŇpp.52, 2015. (in Chinese)

Yang, Q., Song, K., Wen, Z., Hao, X., and Fang, C.: Recent trends of ice phenology for eight large lakes using MODIS products in Northeast China, International Journal of Remote Sensing, 40, 5388-5410, 10.1080/01431161.2019.1579939, 2019.

Sec 2.3: Some comment about the two methodologies adopted are needed, i.e. applicability limits, reliability of results, pros, cons...

Reply: The authors really appreciated the comments. We added some comments on the two methologies, and also explained their application limits, reliability, and pros, and cons as well. Please check the revised manuscript later on will be uploaded when we further did the data analysis, presented the results and discussed these new results.

Figures and caption are generally not consistent. As an example, figure 2 and 3 use both notation DOY and dates, and the label (a) and (b) are never cited in caption.

Reply to comment: The labels (a) and (b) has been cited in the caption of figure 2 and 3.

line 179: PC not yet defined

Reply to comment: The definition has been inserted into the revised manuscript.

Lines 197-200: Figure 5 does not show DOY

Reply to comment: DOY was changed to a specific date, and the caption of figure 5 has been updated, with labels (a) and (b).

Some English revision is needed in the manuscript, for example lines 23-25, 40-43, 215, 299...

Reply to comment: We really appreciate your suggestion. We have carefully revised the manuscript according to the reviewers' comments, and used an English-language editing service Editsprings, to polish our wording.

---

## Author Response (AR1)

**Title: The role of snow on river ice regime across Songhua River basin, Northeast China**

Qian Yang1, 2, Kaishan Song1, Xiaohua Hao3, Zhidan Wen2, Yue Tan1, and Weibang Li1 1Jilin Jianzhu University, Xincheng Road 5088, Changchun 130118 China; E-Mail: jluyangqian10 @hotmail.com

2 Northeast Institute of Geography and Agroecology, Chinese Academy of Sciences, Shengbei Street 4888, Changchun 130102 China; E-Mail: songks@neigae.ac.cn;

3 Northwest Institute of Eco-Environment and Resources, Chinese Academy of Sciences, Donggang West Road 322, Lanzhou 730000, China; E-Mail: haoxh@lzb.ac.cn;

**Response to editor:**

**General comments:**

Your paper received very critical reviews that asked for substantial revisions. Note that the criticism does not only address the presentation of your study but also the contribution of your work to the field.

Thank you for these comments and for the reviewers' comments concerning our manuscript entitled **"The role of snow cover on ice regime across Songhua River basin, Northeast China"** (tc-2019-242). Those comments are all valuable and very helpful for revising and improving our paper, as well as the important guiding significance to our work. We carefully gone through the comments and made extensive corrections accordingly, **marked as red in the manuscript**. Again, please accept the gratitude of all authors from the bottom of the heart and your suggestion enlightened us to think over more deeply than ever before.

**Specific comments**

1) you need to go belong the linear regression analysis of the basic parameters; if a large spatiotemporal dataset is used please consider clustering analysis and/or principal component analysis and/or Bayesian statistics.

Response: Thank you for this helpful suggestion, and we had discussed the possibilities of the three

methods as blow.

We tried to analyze the distribution of complete frozen duration by method of k-means, and it is hard to explain the classification results through topography or climate features, and that's why we did not use clustering analysis.

Principal component analysis (PCA) could be used for two aspect in our work. The distribution of complete frozen duration could be decomposed by PCA, similar to the empirical orthogonal function (EOF) and rotated empirical orthogonal function (REOF). EOF and REOF focused on the eigenvector of original dataset, and PCA focused on the time coefficients, which could reflect the long-term trend of original dataset. The time coverage of our data is only five years, and is suitable for analyzing long-term trend. PCA could also be used for analyzing the relationship between ice regime and impact factors. But our work only considered two factors: air temperature and snow depth. That's why PCA were not used in our work herein.

We used Bayesian linear regression to build the equation between ice thickness and snow depth, air temperature. Two types of air temperature had been considered: the air temperature on bank and the air temperature on bank and the negative cumulative air temperature. Results, snow on ice played a dominant role when the river ice is completely frozen, followed by negative cumulative air temperature. **You can check the changes in Part 2.3.3 and 3.2.2, and we added a new Figure 9 to illustrate the results from Bayesian linear regression**

2) you need to include a literature review describing what is the current state of knowledge in the field and how your study (or objective of your study) advances the current state of knowledge in the field.

**Response:** Thank you for this helpful suggestion, we have updated the introduction as you suggested based on literature review and supplements new references and emphasized on the diversity knowledge on the role of snow cover during the ice process, **seen the line 47-94 of Introduction**.

The surface-based networks, including climatic and hydrological stations, have been established for tracing climate and hydrological changes in Northeast China, which are limited by the accessibility of

surface-based networks and the range of filed measurement. To evaluate the influence of ice regime on regional climate and human environment, a robust investigation and quantitative analysis on ice regime is necessary, which provide helpful information for projecting future changes in the ice regime.

**3) you need to provide a detailed description of the methods used (e.g. data pre and post processing, uncertainty analysis)**

**Response:** The authors really appreciated the comments. We added some comments on the two methods we used, and also explained their application limits, reliability, and pros, and cons as well, seen in Part 2.3 (Line 143 to 185). Besides, we expanded the description of dataset we have used and used sub title to make it clear, seen in Part 2.2 (Line 112 to 141).

**4) you need to expand the discussion and conclusions sections so that they reflect/include all the key results from your analysis**

**Response:** We have significantly improved the conclusion, as you suggested, seen in Line 328 to 350 of Conclusions.

5) you need to pay particular attention to the language and structure of the paper: use clear sentences and logical flow, avoid grammatical and spelling errors, avoid repetitions and redundancy, and definitely proof-read the manuscript before re-submission (I would also recommend giving your manuscript to a native English speaker for proof-reading if this is possible for you)

**Response:**

We really appreciate your suggestion, and we adjusted the structure of this paper. We have carefully revised the manuscript according to the reviewers' comments, and used an English-language editing service Panda Edit Network (http://www.pandaedit.com/), to polish our language and writing styles. The certificate had been uploaded. We provided a comparison between the new version and the previous manuscript.

**The list of improvements.**

| Comments                                                                                                  | Improvements.                                             |
|-----------------------------------------------------------------------------------------------------------|-----------------------------------------------------------|
| 1) you need to go belong the linear regression analysis of the basic parameters; if a large spatio-       | Part 2.3.3 (Line 171 to 185)                              |
| temporal dataset is used please consider clustering analysis and/or principal component analysis and/or   | Part 3.3.2 (Line 298 to 315)                              |
| Bayesian statistics.                                                                                      | Figure 9 (Line 585)                                       |
| 2) you need to include a literature review describing what is the current state of knowledge in the field | Line 47-94 of Introduction.                               |
| and how your study (or objective of your study) advances the current state of knowledge in the field.     |                                                           |
| 3) you need to provide a detailed description of the methods used (e.g. data pre and post processing,     | Part 2.2 (Line 112 to 141).                               |
| uncertainty analysis)                                                                                     | Part 2.3 (Line 143 to 185).                               |
| 4) you need to expand the discussion and conclusions sections so that they reflect/include all the key    | Line 328 to 350 of Conclusions.                           |
| results from your analysis                                                                                |                                                           |
| 5) you need to pay particular attention to the language and structure of the paper: use clear sentences   | Every sentence had been checked. We provided a comparison |
| and logical flow, avoid grammatical and spelling errors, avoid repetitions and redundancy, and            | between the new version and the previous manuscript.      |
| definitely proof-read the manuscript before re-submission (I would also recommend giving your             |                                                           |
| manuscript to a native English speaker for proof-reading if this is possible for you)                     |                                                           |

**Right People, Right Papers**

**Let Scientists Focus on Science!**

**EDITORIAL CERTIFICATE**

This document certifies that the manuscript listed below was edited for proper English language, grammar, punctuation, spelling, and overall style by one or more of the native and subject matter expert English-speaking editors at Panda Edit Network (PEN).

**Manuscript Title**

The role of snow and ice thickness on river ice regime across Songhua River basin, Northeast China

Author Representative

Qian Yang, PhD

**Date Issued**

May 10, 2020

Editor in Chief

Cristina Milesi, PhD

This certificate may be verified at http://www.pandaedit.cn/. Documents receiving this certification are considered English-ready for publication; however, the author is free to further edit the manuscript and reject our edits before submission. To verify the final PEN edited version or if you have any questions or concerns about the edited determent, please contact

**Panda Edit Network: info@pandaedit.com**

We are a team of Professors who wish to provide English language editing for your journal article, grant application, thesis, dissertation and other scientific documents. Unlike other English language editing companies, we are active researchers performing research and publishing in scientific journals just like you. Importantly, we provide and encourage active interaction between the Editors and the Authors through video conferences. This helps Authors understand why the corrections have been made. Our goal is to help improve the Author's writing skills. For more information about our company, services and partner discounts, please visit http://www.pandaedit.cn/.

**Title: The role of snow cover on ice regime across Songhua River Basin, Northeast China**

Qian Yang1, 2, Kaishan Song1,\*, Xiaohua Hao3, Zhidan Wen2, Yue Tan1, and Weibang Li1

1Jilin Jianzhu University, Xincheng Road 5088, Changchun 130118 China; E-Mail: jluyangqian10 5 @hotmail.com

2 Northeast Institute of Geography and Agroecology, Chinese Academy of Sciences, Shengbei Street 4888, Changchun 130102 China; E-Mail: songks@neigae.ac.cn;

3 Northwest Institute of Eco-Environment and Resources, Chinese Academy of Sciences, Donggang West Road 322, Lanzhou 730000, China; E-Mail: haoxh@lzb.ac.cn;

10 Correspondence to: Song K. S. (songks@neigae.ac.cn)

**Abstract:** The Songhua River Basin, located in Northeast China, is an area sensitive to global warming that could be impacted by changes in lake and river ice regimes. The regional role and trends of lake and river ice of this area have been scarcely investigated and are critical for aquatic ecosystems, climate variability, and human activities. Using ice records of local hydrological stations, we examined the spatial variations of the

- 15 ice phenology and ice thickness in the Songhua River Basin from 2010 to 2015 and explored the role of snow depth and air temperature on ice regime. All of five river ice phenology indicators, including freeze-up start, freeze-up end, break-up start, break-up end and complete frozen duration, showed a latitudinal distribution and a changing direction from southeast to northwest. Five typical geographic zones were identified applying a rotated empirical orthogonal function. Maximum ice thickness had a higher correlation with ice phenology,
- 20 especially with the break-up process. Six Bayesian regression models were built between ice thickness, air temperature, and snow depth in three sub-basins of the Songhua River Basin. Results showed significant and positive correlations between snow cover and ice thickness when freshwater was completely frozen. Rather than by air temperature, ice thickness was influenced by negative cumulative air temperature through the heat loss of ice formation and decay.
- 25 **Keywords.** River ice, ice phenology, ice thickness, snow on ice, air temperature, rotated empirical orthogonal function

**1** Introduction**

The freeze-thaw process of surface ice of temperate lakes and rivers plays a crucial role in the interactions among the climate system(Yang et al., 2020), freshwater ecosystems (Kwok and Fahnestock, 1996) and the biological environment (Prowse and Beltaos, 2002). The presence of freshwater ice is closely associated with social and economic activities, ranging from human-made structures, water transportation, to winter recreation (Williams and Stefan, 2006;Lindenschmidt et al., 2017). Ice cover on rivers and lakes exerts large forces due to thermal expansion and could cause extensive infrastructure losses to bridges, docks, and shorelines (Shuter et al., 2012). Ice cover on waterbodies also provides a natural barrier between the

35 atmosphere and water. Ice cover also blocks the solar radiation necessary for photosynthesis to provide

enough dissolved oxygen for fish, thus can have a negative effect on freshwater ecosystems and, in extreme cases, lead to winter kill of fish (Hampton et al., 2017). Generally, the duration of freshwater ice has shown a declining trend, with later freeze-up and earlier break-up throughout the northern hemisphere. For example, freeze-up has been occurring 0.57 days per decade later and break-up 0.63 days per decade earlier during the

- 40 periods of 1846-1995 (Magnuson et al., 2000;Sharma et al., 2019;Beltaos and Prowse, 2009). To evaluate the influence of ice regimes on the regional climate and human environment, and provide helpful information for regional projections of climate and ice-river floods, a robust and quantitative analysis on ice processes is necessary. Despite the growing importance of river ice under global warming, very little work has been undertaken to explain the considerable variation of ice characteristics in Northeast China, where lakes and
- 45 rivers are frozen for as long as five to six months a year.

The earliest ice record in the literature dates back to 150 years ago (Magnuson et al., 2000). Ice development and ice diversity scales have been regarded as sensitive climate indicators. Ice phenology and ice thickness have been studied to gain a deeper understanding of ice processes. At medium and large scales, optical remote

- 50 sensing data are widely used for deriving ice phenology (Song et al., 2014;Šmejkalová et al., 2016), while microwave remote sensing are used to estimate ice thickness and snow depth over ice (Zhang et al., 2019;Kang et al., 2014). Wide-range satellites make it possible to link ice characteristic with climate indices, such as air temperature (Yang et al., 2020) or large-scale teleconnections (Ionita et al., 2018), but their spatial resolutions are too large to detect ice thickness and snow depth accurately at small scales. For example, the
- 55 microwave satellite data of AMSR-E have a spatial resolution of 25 km, but the largest width of Nenjiang River only ranges from 170 to 180 meters. The spatial resolution limits the application of satellite observations to precisely inverse ice thickness, let alone snow depth.
- In terms of point-based measurements, the most commonly used ground observations include regular observations, ice charts, volunteer monitoring and field measurements (Duguay et al., 2015). Ground observations depend on spatial distribution and representation, and are limited by the accessibility of surfacebased networks and the range of field measurement. Ice parameters differ greatly from point to point on a given river (Pavelsky and Smith, 2004), and the uneven distribution of hydrological stations poses an obstacle to gaining a comprehensive understanding of river ice. Various models have been implemented to derive ice
- 65 phenology and ice thickness, such as physically-based models (Park et al., 2016), linear regressions (Palecki and Barry, 1986;Williams and Stefan, 2006), logistic regressions (Yang et al., 2020) and artificial neural networks (Seidou et al., 2006;Zaier et al., 2010). These models consider the energy exchange and physical changes of freshwater ice and require detailed information and data support, including hydrological, meteorological, hydraulic and morphological information. Fixed stations are normally located around the
- 70 river mouth of certain rivers, so these models are limited by the input data available (Pavelsky and Smith, 2004). Both modelling and remote sensing monitoring require sufficient historical ice records to validate and improve accuracy and reliability.

The ice cover of water bodies experiences three stages during which ice phenology, ice thickness and ice composition change greatly. These stages are: freeze-up, ice growth, and break-up (Duguay et al., 2015).

Although air temperature greatly influences the freeze-thaw cycle of river ice, the effect of snow cover can't be ignored. Generally, snow depth outweighs air temperature during the ice forming process and increasing snow depth provides favourable conditions for thicker ice (Morris et al., 2005;Park et al., 2016). Compared to other studies, air temperature had a greater effect on ice thickness than snow depth and were attributed this

- 80 to the high snowfall in the study area (Gao and Stefan, 2004). Besides, in situ observations at Russian river mouths where ice thickness decreased had not shown any significant correlation between ice thickness and snow depth (Shiklomanov and Lammers, 2014). Those studies analysed the relationship in view of spatial distributions and ignored the changing status of ice formation processes. The relative influence of snow depth and air temperature on the ice regime deserves further exploration in Northeast China.
- 85

The surface-based networks, including climatic and hydrological stations, have been established for tracing climate and hydrological changes in Northeast China, which are limited by the accessibility of surface-based networks and the range of filed measurement. To evaluate the influence of ice regime on regional climate and human environment, a robust investigation and quantitative analysis on ice regime is necessary, which

90 provide helpful information for projecting future changes in the ice regime. The previous work explored the ice process in at one or more locations on a given river and ignored the changing regional pattern of ice development due to sparse location. The objectives of this study are to: (1) investigate and compare the spatial distribution of ice phenology and thickness in Northeast China; (2) quantitatively explore the influence of snow cover and air temperature on ice regime.

95

**2 Materials and methods**

**2.1 Study area**

The Songhua River Basin is located in the middle of Northeast China (Figure 1), and includes Jilin Province, Heilongjiang Province, and the eastern part of Inner Mongolia Autonomous Region. The Songhua River is

- 100 the third-longest river in China, and has three main tributaries: Nenjiang River, Main Songhua River, and Second Songhua River (Zhao et al., 2018;Khan et al., 2018). The basins of the three tributary rivers include: Nenjiang Basin (NJ), the Downstream Songhua River Basin (SD), and the Upstream Songhua River Basin (SU) (Figure 1). The Nenjiang River has a length of 1370 km, and the corresponding drainage has an area of 2.55 × 106 thousand km2; the Main Songhua River has a length of 939 km and the downstream catchment of
- 105 the Songhua River Basin (SD) has an area of  $1.86 \times 10^6$  km2; the Second Songhua River has a length of 958 km and the upstream catchment of the Songhua River Basin (SU) has an area of  $6.19 \times 10^6$  km2 (Yang et al., 2018;Chen et al., 2019). The whole Songhua River Basin is characterized by temperate and cold temperate climates: winter is long and cold; spring is windy and dry. Annual average air temperature ranges between 3 to 5°C, while annual precipitation ranges from 400 to 800 cm from the southeast to the northwest. (Wang et
- 110 al., 2015;Wang et al., 2018).

**[Figure 1 is added here]**

**2.2 Data Source**

**2.2.1 Ice phenology**

The hydrographic bureau of the Chinese Ministry of Water Resources has established a remarkable observation network for ice regimes. The ice records of the Songhua River Basin were obtained from the annual hydrological report, including ice phenology, ice thickness, snow depth on ice and air temperature on bank (BAT) (Annual hydrological report, 2010-2015). To analyse the spatial pattern of the ice regime, we explored five river ice parameters with the corresponding day of year (DOY) from 158 stations. We located 50, 36 and 72 stations in the NJ, SU and SD basins, respectively. For each record, five lake ice phenological

- 120 events were derived from the annual hydrological report; the definitions referred to specification for observation of ice regimes in rivers and previous works (Cai et al., 2019;Yang et al., 2019;Duguay et al., 2015):
  - Freeze-up start (FUS) is considered the first day when floating ice can be observed with temperatures below 0 °C;
- 125 Freeze-up end (FUE) is the day when a steady ice carapace can be observed on the river, and the area of ice cover is more than 80% in the view range;
  - Break-up start (BUS) is the first day when ice melting can be observed with surface ponding;
  - Break-up end (BUE) is the day when the surface is mainly covered by open water and the area of open water exceed 20%;
- 130 Complete frozen duration (CFD) is the ice cover duration when the lake is completely frozen during the winter, from FUE to BUS.

**2.2.2 Ice thickness**

To study seasonal changes in ice thickness (IT) and establish the regression model, we used ice thickness, snow depth and air temperature from 120 stations for the period ranging from 2010 to 2015. We used 37, 28

- 135 and 55 stations located in the NJ, SU and SD basins, respectively. The hydrological report provided ice thickness, snow depth on ice and BAT every five days from November through April, totalling 37 measurements in one cold season. The yearly maximum ice thickness (MIT) of the river centre and the corresponding DOY were calculated from five-day records. The average snow depth (ASD) was calculated from the mean of three or four measurements around the ice hole for ice thickness measurement without
- 140 human disturbance. To enhance the performance of the regression model, negative cumulative air temperature was calculated from air temperature from November to March.

**2.3 Data analysis**

**2.3.1 Kriging**

Kriging has been widely used to spatially interpolate in situ measurements of ice phenology to understand its
spatial distribution (Choiński et al., 2015;Jenson et al., 2007). Kriging assumes a correlation between regionalized variables and variograms that reflects randomization and structuredness of regionalized variables. It estimates unknown values based on the best linear unbiased estimator with minimal variance, expressed as:

$$\hat{Z}(s_o) = \sum_{i=1}^N \lambda_i Z(s_i)$$

- 150 where  $\hat{Z}(s_o)$  is the estimate by kriging at an unknown point  $s_o$ ,  $Z(s_i)$  is the variable at a measured point  $s_i$ , N is the number of measured points.  $\lambda_i$  is a weight for  $Z(s_i)$ , and relies on the spatial arrangement of the measured values and the distance between the prediction location and the measured location (C.R. Paramasivam, 2019). The average values of five ice phenology indicators during the six years were interpolated to create isophenes, i.e., contour lines connecting locations with the same ice phenology.
- 155

**2.3.2 Rotated empirical orthogonal function (REOF)**

Empirical orthogonal function (EOF) decomposition is commonly used in climate and hydrological analyses (Bian et al., 2019; Yang et al., 2017). Its basic principle is to decompose the field containing p spatial points (variable) over time. If the sample size is n, then the data value  $x_{ij}$  including specific spatial point i and specific

160 time *j* in the field can be regarded as the linear combination of spatial modes and temporal modes according to equation:

$$S_{b^2}^2 = \frac{1}{mM} \sum_{j=1}^{M} \sum_{p=1}^{m} (b_{jp}^2 - \overline{b^2})^2$$

where  $b_{jp}$  is the *j* th loading coefficient of the *p* th EOF mode.

- The major advantages of the EOF method is to separate the uncorrelated components that confuse the spatial information and make it hard to interpret a physical phenomenon. In order to solve these problems, a rotated EOF (REOF) rotates the original EOF matrix into a new matrix in which the squared elements of the eigenvectors are maximum, which can better reflect changes across different geographic regions and identify correlations. This paper presented the first four load vectors of the CFD decomposed by REOF and their corresponding principal components (PC) to identify the typical geographic zones in Northeast China.
- 170

**2.3.3 Bayesian linear regression**

Ice thickness had been modelled by air temperature and snow depth using Bayesian linear regression (BLR), which has been widely used in hydrological and environmental analyses (Zhao et al., 2013;Gao et al., 2014). BLR treats regression coefficients and the disturbance variance as random variables, rather than fixed but

- 175 unknown quantities. This assumption leads to a more flexible model and intuitive inferences (Barber, 2008). The BLR model was implemented through two models: a prior probability model considered the probability distribution of the regression coefficients and the disturbance; a posterior model predicted the response using the prior probability mentioned below. The performance of the regression model was evaluated using the determination coefficient R2 and the root mean square error (RMSE). In this paper, the Y data were the five-
- 180 day ice thickness values, and the X data included snow depth over ice and air temperature on the river bank. The calculation of the regression used the in-situ measurements from November to March and excluded the ice records of April due to unsteady ice conditions. Two types of air temperature were considered: BAT and negative cumulative air temperature (ATC). Additionally, the Pearson correlation was calculated to analyse the relationship among the five ice phenology events and ice-related parameters, including MIT, ASD, and

185 BAT(Gao and Stefan, 1999; Williams et al., 2004).

**3** Results and discussion**

**3.1 Spatial variations of river ice phenology**

**3.1.1 Freeze-up and break-up process**

Figure 2 illustrates the average spatial distribution of FUS and FUE interpolated by kriging and the isophenes
in the Songhua River Basin of Northeast China from 2010 to 2015. Figure 3 illustrates the spatial distribution of the BUS and BUE. The corresponding statistics are listed in Table 1. FUS ranged from October 28th to November 21st with a mean value of November 7th, and FUE ranged from November 7th to December 8th with a mean value of November 22nd. BUS ranged from March 24th to April 20th with a mean value of April 9th, and BUE ranged from March 31th to April 27th with a mean value of April 15th. These four parameters showed

195 a latitudinal gradient: FUS and FUE decreased while BUS and BUE increased as the latitude increased, except in NJ. The middle part of NJ had the highest FUS and FUE and decreased to the southern and northern part. As the latitude decreased, the air temperature tended to increase, leading to later freeze-up and earlier breakup with shorter ice-covered duration; vice versa.

[Figure 2 is added here]

[Figure 3 is added here]

[Table 1 is added here]

**3.1.2 Complete frozen duration**

Figure 4(a) illustrates the average spatial distribution of CFD interpolated by kriging and the isophenes in the Songhua River Basin from 2010 to 2015. CFD ranged from 110.74 to 163.00 days with a mean value of 137.86 days, increasing with latitude. Interestingly, the isophenes of CFD had different directionality, increasing from the southeast to northwest, which could also be found in the other four ice phenologists. Both FUS and FUE correlated negatively with latitude, with coefficients of -0.66 and -0.53, respectively (n=158, p < 0.001). BUS, BUE and CFD were all positively correlated with latitude with coefficients of 0.48, 0.57 and 0.55, respectively (n=158, p < 0.001). High values indicated a delay in the ice phenology event. The</li>

210 general spatial trend was a tendency to advance as the latitude increased for the FUS and FUE, a tendency for delay for BUS and BUE, and a lengthening tendency for CFD. A decreasing solar radiation with latitude could explain this trend, which is directly connected with the ice thaw and melting processes.

To find the spatial distribution of ice durations, average values of CFD between 2010 and 2015 were decomposed by REOF, and the spatial distribution of the first four PC are shown in Figures 4 (c)-(f) interpolated by kriging. The first to fourth PC modes accounted for 45.89%, 13.22%, 12.62%, and 12.00%, respectively, with the cumulative variance of 83.73%. The PC data ranged from -0.22 to 0.15, and the areas with high values presented a planar distribution, which were further regarded as five typical geographic zones considering the topography of Northeast China. Zone 1, located in the Three River Plain, where Heilongjiang,

220 Wusuli, and Songhua River converge together, was identified from the first PC. Zone 2, located around

Heaven Lake of Changbai Mountain, in the southernmost part and which has the highest elevation of 2565 m, was identified from the second PC mode. We excluded a planar distribution above Zone 2 because of the gentle terrain in the Songhua River Basin. The middle part of the Songhua River Basin accounts for a large area where no typical zones were found. The REOF was good at enhancing the high-value areas, and the PC

- 225 data of this area around 0 were ignored. Zone 3, located on the eastern edge of the three basins with relatively high elevation along the ridge of Changbai Mountain, was identified from the third PC mode. Based on the fourth PC mode, Zone 4 was determined in the northernmost part along the ridge of Xiao Higgan Mountain where it meets with Da Higgan Mountain. Zone 5 almost covered the southern part of the NJ basin along the ridge of Da Higgan Mountain and appeared in the second, third, and fourth PC. The final distribution was
- 230 identified from the convergence area of these three modes.

[Figure 4 is added here]

**3.2 Variations of ice thickness**

**3.2.1 Spatial pattern of ice thickness**

- Figure 5 illustrates the spatial distribution of the yearly maximum ice thickness (MIT) of the river centre and the corresponding DOY. Table 2 summarized the statistical result of MIT and DOY. MIT ranged from 12 cm to 146 cm, with an average value of 78 cm. The MIT between 76 and 100 cm accounted for the largest percentage of 43.33%, followed by 31.67% of MIT between 50 and 75 cm. Five stations had MIT greater than 125 cm. Two stations were located in Zone 3 and three stations in Zone 4. The DOY of MIT had an average value of February 21st, and MIT mainly occurred 59 and 40 times in February and March, respectively.
- 240 Four of the five highest MITs greater than 125 cm happened in March, which is consistent with the interannual changes in ice development shown in Figure 6. The results suggested that the river ice is always thickest and most steady in February or March, which is the best suitable time for human activities such as ice fishing and entertainment. The ice thickness didn't show the same latitudinal distribution as ice phenology, which suggested that more climate factor should be taken in to consideration, such as snow depth and wind.
- 245

[Figure 5 is added here]

[Table 2 is added here]

**3.2.2 Seasonal changes of ice thickness**

Figure 6 displays the seasonal changes of ice development using ice thickness, average snow depth on ice,
and BAT every five days from 2010 to 2015. Among the three basins, NJ had the highest snow depth of 29.15 ± 9.99°C, followed by -25.61 ± 9.02 °C in the SD, and -22.17 ± 7.33 cm in the SU. SD had the highest snow depth of 9.18 cm ± 3.39 cm on average level, followed by 8.35 cm ± 4.60 cm in SU, and 8.23cm ± 3.92 cm in NJ. The changes in IT and ASD had similar overall trend, while BAT followed the opposite trend. Both IT and ASD increased from November and reached the peak in March, then dropped at the beginning of April.

255 The ASD showed an obvious trend and reached the bottom in the middle of January, which is earlier than the

peaks of MIT and ASD. The NJ and the SD basins underwent greater fluctuations than the SU basin, because river ice may freeze and thaw alternatively at relatively low temperatures. The changes of ice characteristics differed greatly due to time and location; an analysis of the annual changes was not conducted because the time series were not long enough.

**260 [Figure 6 is added here]**

**3.3 The relationship between ice regime and climate factors**

**3.3.1 Correlation analysis**

Figure 7 displays the correlation matrix between lake ice phenology events and three parameters, covering yearly average values of ASD, BAT, and MIT with a dataset size of 120 stations. Colour intensity and sizes of the ellipses are proportional to the correlation coefficients. MIT had a higher correlation with four of the five indices than ASD and BAT, except with FUS, with which both MIT and BUE had the highest correlation of 0.63 (p<0.01, n=120). During the freeze-up process, two freeze-up dates had a negative correlation with MIT and ASD; during the break-up, two break-up dates had a positive correlation with MIT and ASD. The situation of BAT was contrary to that of MIT and ASD.

270 Regarding to the annual changes, no significant correlation was found between snow depth and five ice phenology events in Figure 7.

**[Figure 7 is added here]**

Figure 8 shows the bivariate scatter plots between yearly maximum ice thickness (MIT) and five ice phenology indicators with regression equations. The break-up process had a negative correlation with MIT,

- while freeze-up had a positive correlation. Besides, the break-up process had a higher correlation with MIT, and BUS had the highest correlation coefficients with MIT of 0.65 (p<0.01). CFD also had a positive correlation with MIT of 0.55 (P<0.01), which means that a thicker ice cover in winter leads to a delay in melting time in spring. The break-up not only depends on the spring climate conditions, but is also influenced by ice thickness during last winter. A thicker ice cover stores more heat in winter, taking a longer time to melt in spring. The limited performance of the regression model could be attributed to the difficulties in
- 280 melt in spring. The limited performance of the regression model could be attributed to the difficulties in determining river ice phenology. Although a uniform observation protocol was required, the repaid transition between frozen river and open water for two or three days with floating ice and the inhomogeneities among different stations could not be ignored.

**[Figure 8 is added here]**

285 To further explore the role of snow cover, the monthly correlation coefficients between IT and ASD, and IT and BAT were calculated and listed in Table 3. The correlation coefficients between IT and ASD increased from November to March and reached a peak of 0.75 in March when ice was thickest. This indicated an increasingly important role of snow depth on ice thickness as the ice accumulated. The higher correlation coefficients between IT and BAT in November and December revealed that BAT played a more important

- role in the freeze-up process. Moreover, whether the status of river ice was steady or not also could not be neglected when studying the role of snow cover.
  The positive correlation coefficient between snow depth and ice thickness (Table 3) revealed two opposite effects of snow depth during ice development: during the ice-growth process, snow depth protects the ice from cold air and slows down the growth rate of ice thickness; during the ice-decay process, the lake bottom
- 295 ice stops to grow, and the snow mixes with surface ice into slush and promotes melting.

**[Table 3 is added here]**

**3.3.2 Regression modelling**

Figure 9 illustrates the scatter plot between measured and predicted ice thickness using Bayesian linear regression in three sub-basins in Northeast China. R2 ranged from 0.81 to 0.95, and RMSE ranged from 0.08
to 0.17. The model worked best in the SU basin, followed by the NJ and the SD basins. Figure 9 indicates that snow depth outweighed air temperature in terms of effect on ice thickness, which is consistent with previous studies. Moreover, replacing BAT with ATC enhanced the model performance in all three basins, revealing a more important role of ATC than BAT.

**[Figure 9 is added here]**

- 305 The correlation between air temperature and ice regime was not as significant as in previous studies for several reasons. Average air temperatures were most commonly calculated over fixed time periods at regional scales, for example as moving averages for certain time periods. The seasonal changes of air temperature were ignored, as well as their effects within one cold season. The negative ATC behaved better than BAT when building the Bayesian regression equation, which suggested that heat exchanges between river surface
- 310 and atmosphere dominated the ice process. Heat loss is mainly made up of sensible and latent heat exchange, which is proportional to negative ATC during the cooling process. During the complete frozen duration, snow depth along with wind speed began to influence the heat exchange and ice thickening. Air temperature exerted a lesser effect on spring break-up, which is more dependent on the ice thickness and snow depth. In summary, snow depth dominated the ice process when the river was completely frozen, while cumulative air temperature
- 315 dominated during the transition process.

**4** Conclusions**

Five river ice phenology indicators, including FUE, FUS, BUE, BUS, and CFD, in the Songhua River Basin of Northeast China have been investigated using in situ measurements for the period 2010 to 2015 using kriging and REOF methods. The FUS and FUE decreased while the BUS, BUE, and CFD increased with
latitude. The five river ice phenology indicators followed the latitudinal gradient and a changing direction from southeast to northwest. The highest MIT over 125 cm were distributed along the ridge of Da Hagan Lin and Changbai Mountain, and MIT occurred most often in February and March, which indicated that this is the safest period for human activities such as navigation and winter recreation. Five typical geographic zones were identified from the first four PC modes of CFD, covering Changbai Mountain, Three River Plain, Da

325 Higgan Mountain, and Xiao Higgan Mountain, providing a deeper understanding of river ice distribution and its relationship with geographic locations and topography in Northeast China.

Within one cold season, ice thickness and snow depth showed similar seasonal changes, i.e. first increased and then decreased, while air temperature showed an opposite trend. The peaks of snow depth and ice thickness fell behind air temperature for almost one month. High correlation coefficients between yearly

- maximum ice thickness and ice phenology indicators revealed that ice phenology is closely related to ice thickness, especially in the break-up process. The yearly analysis failed to explain the relationship between ice regime and snow depth and air temperature. Based on monthly correlation analysis, snow cover played an increasingly important role as the ice cover become steady. Additionally, air temperature associated with
- ice phenology more closely than ice thickness.

330

340

Six Bayesian regression models were built between ice thickness and air temperature and snow depth in three sub-basins of Songhua River, considering two types of air temperature: air temperature on bank and negative cumulative air temperature. Results showed that snow cover correlated with ice thickness significantly and positively during the periods when the freshwater was completely frozen, and negative cumulative air temperature influenced the thickness rather than air temperature through the heat loss of ice formation and

- decay. The negative ATC behaved better than BAT when building the Bayesian regression equation, which suggested that heat exchanges between the river surface and the atmosphere dominated the ice process.
- 345 This study aimed at exploring the regional patterns of river ice development based on in situ measurements and was limited by data accessibility. Remote sensing data could provide long-term and wide-range information for ice thickness and ice phenology since 1980, expanding our study scope. The work herein will provide a valuable reference for the retrieval of ice development by remote sensing. Knowing the long-term change of river ice and the future projection could provide information for evaluating the influence of climate
- 350 on social-economics, ecological environment and human activists across the riparian zones.

**Abbreviations**

The following abbreviations are used in this manuscript:

AMSR-E Advanced Microwave Scanning Radiometer- Earth Observing System

- ASD Average Snow depth
- 355 ATC Cumulative air temperature
  - BAT Air temperature on bank
  - BLR Bayesian linear regression
  - BUS Break-up start
  - BUE Break-up end
- 360 CFD Completely frozen duration
  - DOY Day of year
  - EOF Empirical orthogonal function

- FUS Freeze-up start
- FUE Freeze-up end
- 365 IP Ice phenology
  - IT Ice thickness
  - NJ Nenjiang River Basin
  - MIT Maximum ice thickness
  - PC Principal component
- 370 REOF Rotated empirical orthogonal function
  - RMSE Root mean square error
  - SD Downstream Songhua River Basin
  - SRTM Shuttle Radar Topography Mission
  - SU Upstream Songhua River Basin

**375 Author Contribution**

Song K.S. and Yang Q. designed and conducted the idea of this study. Yang Q. Wen Z.D. wrote the paper and analysed the data cooperatively; Hao X.H. provided value suggestion for the structure of study and paper; Li W.B. and Tan Y. exerted efforts on data processing and graphing. This article is a result of collaboration with all listed co-authors.

**380 Competing interest**

The authors reported no potential conflict of interest.

**Acknowledgments**

The research was sponsored by the National Natural Science Foundation of China (41801283, 41971325, 41730104). The anonymous reviewers to improve the quality of this manuscript are greatly appreciated.

**Tables**

Table 1 Summary statistics of ice phenology interpolated by Kriging from 2010 to 2015. The ice phenology indicators included freeze-up start (FUS), freeze-up end (FUE), break-up start (BUS), break-up end (BUE), complete frozen duration (CFD). NJ, SD and SU represent the Nenjiang Basin, the
downstream Songhua River Basin (SD) and the upstream Songhua River Basin (SU). DOY denotes day

| р . | G4 41 41   | FUS    | FUE    | BUS    | BUE    | CFD    |
|------------|------------|--------|--------|--------|--------|--------|
| Basins     | Statistics | (DOY)  | (DOY)  | (DOY)  | (DOY)  | (day)  |
|            | Maximum    | 319.14 | 334.98 | 110.54 | 117.61 | 163.00 |
| NT         | Mean       | 307.02 | 324.58 | 98.65  | 106.64 | 139.39 |
| INJ        | Minimum    | 301.41 | 311.30 | 84.53  | 90.40  | 119.11 |
|            | Std Dev.   | 3.91   | 5.69   | 8.16   | 6.80   | 13.22  |
|            | Maximum    | 321.08 | 334.36 | 110.01 | 102.84 | 154.06 |
| CD         | Mean       | 313.74 | 326.70 | 102.55 | 97.15  | 140.86 |
| 5D  | Minimum    | 305.64 | 316.80 | 93.22  | 92.37  | 125.32 |
|            | Std Dev.   | 2.83   | 3.13   | 3.92   | 2.12   | 5.69   |
|            | Maximum    | 325.92 | 342.09 | 98.25  | 114.37 | 133.62 |
| SU         | Mean       | 320.39 | 334.35 | 91.93  | 106.43 | 122.61 |
| 50         | Minimum    | 313.79 | 327.68 | 83.46  | 95.69  | 110.74 |
|            | Std Dev.   | 2.34   | 3.09   | 3.21   | 4.24   | 4.85   |
|            | Maximum    | 325.92 | 342.09 | 110.54 | 117.61 | 163.00 |
| Tatal      | Mean       | 311.16 | 326.58 | 99.25  | 105.38 | 137.86 |
| 1 otal     | Minimum    | 301.41 | 311.30 | 83.46  | 90.40  | 110.74 |

of year. Std Dev. denotes standard deviation.

Std Dev.

530

5.74

Table 2 The Frequency of yearly maximum ice thickness from November to April. The row represents different year in cold season and the column represents yearly maximum ice thickness with the unit of cm.

5.54

7.17

6.34

11.68

| MIT
Month |

---

## Author Response (AR2)

**Title: The role of snow cover on ice regime across Songhua River Basin, Northeast China (tc-2019-242)**

Qian Yang[1, 2], Kaishan Song[1, *], Xiaohua Hao[3], Zhidan Wen[2], Yue Tan[1], and Weibang Li[1]

[1] Jilin Jianzhu University, Xincheng Road 5088, Changchun 130118 China; E-Mail: jluyangqian10 @hotmail.com

[2] Northeast Institute of Geography and Agroecology, Chinese Academy of Sciences, Shengbei Street 4888, Changchun 130102 China; E-Mail: songks@neigae.ac.cn;

[3] Northwest Institute of Eco-Environment and Resources, Chinese Academy of Sciences, Donggang West Road 322, Lanzhou 730000, China; E-Mail: haoxh@lzb.ac.cn;

Correspondence to: Song K. S. (songks@neigae.ac.cn)

**General comments**

Dear authors,

I received the comments from referees on your revised draft. Despite your extensive revisions, one of the referees is still not convinced by the contributions of your study to the field. Please try to be clear in your discussion and conclusion how your study advances the current knowledge in the field.

Furthermore, as indicated by the referee, there is an ambiguity between between lake and river ice throughout the manuscript to the point that it is not clear which one you are reporting on (is it river ice characteristics or lake ice?) These are fundamentally two different processes of ice formation and decay. Please try to address these issues carefully.

The objectives of each data processing step are not clearly outlined. Some parameters seem to have unusual naming (such as Negative Cumulative Air Temperature which is referred to as cumulative degree day of freezing in the literature). Please chose a correct (commonly used) terminology and stick with it throughout the text.

Furthermore, there are still typos and redundancy in the writing (see my specific comments).

In the light of this criticism I have reviewed the revised version myself. Please find below my specific comments. I am asking you to address the comments (more general

comments above and all of my specific comment below) and submit the revised manuscript with line-by-line responses to the comments.

Best regards,

Valentina

**Response to General comments**

Dear Valentina,

Thank you for your letter and for the reviewers' comments concerning our manuscript entitled "The role of snow cover on ice regime across Songhua River Basin, Northeast China" (tc-2019-242). Those comments are all valuable and very helpful for revising and improving our paper, as well as the important guiding significance to our work. We carefully gone through the comments and made corrections accordingly, marked as red in the manuscript.

Firstly, we presented a new geographic map of Songjiang River Basin, and supplemented the distribution of rivers in our study area. The figure clearly illustrated the hydrological stations are located on the rivers not on the lakes. Secondly, the data processing had been rewriting and linked with the objectives of our work, and you can checke in the new version of manuscript. Thirdly, we checked and updated the terminology throughout the manuscript, and removing the abbreviations. Besides, we checked typos and redundancy and modified accordingly. Last but not least, please accept the gratitude from the bottom of our heart for your extensive efforts.

Best wishes,

Qian Yang

On behalf of all authors.

**Response to specific comments:**

1. Abstract: 'Five typical geographic zones weridentified applying a rotated empirical orthogonal function.' -> It's not clear what is meant by five typical geographical zones. Typical of what? Also, to what data is the EOF applied?

**Reply to comments:** Thank you for this insightful suggestion, and we considered it carefully. In the previous version of the manuscript, the typical geographical zones were identified visually from EOF results of complete frozen duration. Our original purpose of typical geographic zones is to present a classification suitable for both ice thickness and ice phenology. Considering the limitation of the method, EOF is not a proper and reasonable method for the classification of geographic zones. Thus, we abandoned this geographic zoning method in the current version of the manuscript.

Further, we performed grouping analysis for complete frozen duration in ArcGIS software, as shown in Figure 1. We selected K means as the spatial constraints and MANHATTAN as distance method to classify the completely frozen duration. We tested the group number ranging from 2 to 15, and Figure 1 (d) illustrated the Calinski-Harabasz pseudo F-statistic as the group number increase from 2 to 15. The larger F-statistic reveals how grouping result will be more effective at distinguishing the features and variables. The group number of 2, 3 and 4 have the three most largest F-statistic of 68.24, 65.7 and 63.36. Considering the basin boundaries and topography in the Songhua River Basin, group number of 4 are the best choice for classification.

[Figure]

Figure 1 The grouping analysis result of complete frozen duration. (a), (b) and (c) display the spatial distribution of grouping with group number of 2, 3 and 4 namely. (d) listed Calinski-Harabasz pseudo F-statistic as the group number increase from 2 to 15.

We also performed grouping analysis for maximum ice thickness, and Figure 2 illustrated the Calinski-Harabasz pseudo F-statistic as the group number increase from 2 to 15. The F-statistic increased as the group number increase, which makes it hard to identify the best group number. Besides, the global Moran's index of maximum ice thickness is 0.01 with z scores of 0.13 and p value of 0.89, which indicated that no cluster pattern existed for maximum ice thickness at the 95% confidence level. Therefore, we concluded that ice thickness didn't exhibit the similar grouping cluster as complete frozen duration.

[Figure]

Figure 2 The Calinski-Harabasz pseudo F-statistic of grouping analysis of maximum ice thickness as the group number increase from 2 to 15.

2. Section 2.3.1: Please include more details on what data is used here, i.e. to which data is kriging applied to and what is the specific goal (in your study) that you try to achieve with the kriging. As it currently ready, the section generally describes kriging, but necessary specifics for your study/data are not given.

**Reply to comments:** Thank you for pointing this out, and we have improved the description of Kriging. Kriging has been widely used to spatially interpolate in situ measurements of ice phenology to understand its spatial distribution (Choiński et al., 2015; Jenson et al., 2007). The average values of five ice phenology were calculated during the periods from 2010 to 2015 and explored in the Geostatistical wizard of ArcGIS software, and the interpolation results exhibited their spatial distribution. We chose the ordinary Kriging method and set variation function as the spherical model. Moreover, isophenes connecting locations with the same ice phenology were also graphed based on interpolation results (C.R. Paramasivam, 2019). (Page 6, Line 169-176)

[Figure]

Figure 3 The distribution of hydrological stations, rivers, and lakes in the Songhua River Basin of Northeast China.

18. I would suggest removing the abbreviations here (FUS, FUE, BUS, BUE, CFD, IT, MIT, ASD) as there are too many abbreviations in the study and it is hard to keep track.

Too many abbreviations can disrupt the narrative. If you are using these abbreviations in the figures, then introduce them in the figure captions.

**Reply to comments:** We feel sorry for the improper usage of abbreviations. In our resubmitted manuscript, the abbreviations is removed. Thanks for your correction.

19. Page 4, Line 133: 'establish the regression model' -> it is not clear what regression model you will be establishing. Will this regression model be described later? If yes, then say 'establish a regression model, which we describe below.'

**Reply to comments:** Thank you for this helpful suggestion, and we modified as you suggested. (Page 5, Line 149-150)

20. Section 2.3 I suggest to provide here a short summary of overall methodology, so the readers can get a general idea of what is to follow and how it fits together. So here you would mention the key steps you will use (krigging, EOF and regression) outlining the main objective of each step. Then in each section that follows you provide more details on each step (as you have already done). Without this summary, it is very hard to comprehend why you chose to use each of this methods.

**Reply to comments:** Thank you for this helpful suggestion, and we added a paragraph. To start with, we analyzed the spatial pattern and mapped the cluster of rive ice phenology using Kriging, Moran's index. Then, we explored the relationship between the ice regime and the impact factors using correlation analysis. Last but not least, we established the quantitative relationship between ice thickness and snow depth, air temperature based on the Bayesian linear regression. (Page 6, Line 161-165)

21. Section 2.3.1: Please include more details on what data is used here, i.e. to which data is krigging applied to and what is the specific goal (in your study) that you try to achieve with the krigging. As it currently ready, the section generally describes krigging, but necessary specifics for your study/data are not given.

'The average values of five ice phenology indicators' -> please specify the indicators and specify how the averaging is performed (in time or space or both?)

**Reply to comments:** Thank you for the question, and we made it clear. For each hydrological station, the average values of five river ice phenology were calculated from the ice records from 2010 to 2015 and interpolated by Kriging method to analyze the spatial distribution of river ice phenology. The river ice phenology included freeze-up start, freeze-up end, break-up start, break-up end and completely frozen duration. (Page 6, Line 168-175)

22. Section 2.3.1: Same comment as above. Please explain why EOF is used here (what are you trying to achieve in your study) and to what data EOF method is applied.

'The major advantages of the EOF method is to separate the uncorrelated components that confuse the spatial information and make it hard to interpret a physical phenomenon.' -> you speak about the advantages of EOF which you then immediately present as problematic and therefore use the rotated EOF to solve those problems. So the section here is somewhat confusing. Please revise so that it is clear: what the advantages of rotated EOF are, and why for your specific case you chose to use rotated EOF rather than EOF.

**Reply to comments:** Thank you for the concerns and comments. The typical geographical zones were identified visually from EOF results of completely frozen duration. As mentions before, EOF is not a proper and reasonable method for classification in our case study. After the trial of several grouping methods, we decide using the pattern explored by Moran index, and we explained the further work for exploring spatial pattern above. (Page 6, Line 177-185)

23. Page 5: Please reduce the use of abbreviations. For example, remove BLR and ATC. Results and Discussion: due to too many abbreviations it is hard to follow the narrative. I hope this will improve with the removal of abbreviations and using the full words.

**Reply to comments:** Sorry for this problem, and we have removed all the abbreviations.

24. Page 5, ln 180: 'the Y data were the five day ice thickness values, and the X data

included snow depth over ice and air temperature on the river bank. -> is the Y data the ice thickness values on rivers or lakes? Please clarify. Also, is Y-data presented as five day moving average throughout the whole observational period? Please clarify. For many observations (points) do you have in the regression model? Do you apply any cross-validation, i.e. using section of the data for the model calibration and then another section (independent of calibration data) for validation? If you don't use any cross-validation, how do you prevent over-fitting to noise?

**Reply to comments:** Thank you for pointing this out, and we clarified these questions. In this paper, we treated ice thickness as the Y data values on the riverbank, and the snow depth and air temperature as X data with dataset size of 31. The ice thickness was measured on the riverbank every five days from November to March when the river was completely covered with ice with air temperature below $0\,℃$. Y data is not the five-day moving average.

We carried out cross-validation for Bayesian linear regression using k-fold method and set K value as 5. For each iteration, a different fold is held-out for testing, and the remaining 4 subsets is used for training. The training and testing were repeated for 5 iterations, and Table 1 listed the $R^2$ of the training and testing process each iteration. The best Bayesian linear regression was determined when the bias between testing and training regression was smallest. (Page 7, Line 196-199; Page 12, Line 354-359)

Table 1 The cross-validation of Bayesian linear regression using k-fold method. The $R^2$ values of training dataset and testing dataset based on the Bayesian regression. Ice thickness was treated as dependent variables, and air temperature, snow depth on ice as independent variables. Air temperature and cumulative air temperature of freezing were considered in the model built.

| Basin | Air temperature | | Cumulative air temperature | |
|---|---|---|---|---|
| | Training | Testing | Training | Testing |
| NJ | 0.80 | 0.99 | 0.84 | 0.99 |
| | 0.89 | 0.80 | 0.90 | 0.86 |
| | 0.84 | 0.92 | 0.89 | 0.82 |
| | 0.90 | 0.56 | 0.91 | 0.61 |
| | 0.85 | 0.91 | 0.89 | 0.89 |
| SU | 0.83 | 0.92 | 0.95 | 0.98 |
| | 0.83 | 0.65 | 0.96 | 0.83 |
| | 0.81 | 0.94 | 0.95 | 0.99 |
| | 0.84 | 0.79 | 0.95 | 0.93 |
| | 0.82 | 0.82 | 0.94 | 0.98 |
| SD | 0.80 | 0.96 | 0.82 | 0.98 |
| | 0.84 | 0.16 | 0.86 | 0.25 |
| | 0.81 | 0.84 | 0.82 | 0.87 |
| | 0.79 | 0.97 | 0.79 | 0.96 |
| | 0.81 | 0.80 | 0.82 | 0.83 |

25. Results and Discussion: due to too many abbreviations it is hard to follow the narrative. I hope this will improve with the removal of abbreviations and using the full words.

**Reply to comments:** Sorry again for this problem, and we have removed all the abbreviations. Thanks for your correction. (Page 20, Line 590-595)

26. Page 6, line 210: 'The general spatial trend was a tendency to advance as the latitude increased for the FUS and FUE...' Not clear what is meant by 'a tendency to advance', advance in what? Please revise.

**Reply to comments:** Thank you for this useful suggestion. We built the linear regression equation between the five river ice phenology and latitude. AS the latitude increased by one degree, freeze-up start and freeze-up end happened 2.56 and 2.32 days early, the break-up start and break-up end arrived 2.36 and 2.37 days late, resulting in 4.48 days decrease in completely frozen duration. This could be explained by the decreasing solar radiation with latitude influencing the ice thaw and melting processes

directly. (Page 8, Line 246-251)

27. Page 6: Results from EOS -> it seems that you are visually clustering the PC results (on the map). Is that correct? If yes, please state so. Currently it reads as if you are visually identifying the modes of PC, while these are actually derived from the EOF method itself.

**Reply to comments:** Thank you for this useful suggestion. We found that the results obtained with REOF were probably not appropriate, and it is difficult to explain the distribution characteristics and identified the typical zones using REOF methods. Our further work has been described above.

28. Page 8, ln 259: 'an analysis of the annual changes was not conducted because the time series were not long enough' -> what is considered long enough here?

**Reply to comments:** We thank the reviewer for his/her insightful comments, and we removed the sentence.

29. Page 8, ln 263: 'matrix between lake ice phenology' -> what variables are use for the ice phenology? Please clarify this already in the methods section.

**Reply to comments:** Thank you for this helpful suggestion, and we clarified these parameters as follows:

Figure 7 displays the correlation matrix between lake ice phenology and three ground measurements with a dataset size of 120 stations. The lake ice phenology included freeze-up start, freeze-up end, break-up start, break-up end and complete frozen duration. The three ground measurements included yearly mean values of snow depth, air temperature on bank, and maximum ice thickness. (Page 10, Line 305-309)

30. Page 8, ln 281: 'Although a uniform observation protocol was required, the repaid transition between frozen river and open water for two or three days with floating ice and the inhomogeneities among different stations could not be ignored.' The use of terminology here, such as 'protocol' and 'repaid', does not seem right. Please revise with

wording that is appropriate for physical processes. Start by simplifying the sentence. What do you actually try to say here?

**Reply to comments:** Thanks for your insightful suggestions. The sentence is hard to understand, and we kept part information: Although a uniform specification for ice regime observations was required, the inhomogeneities among different stations could not be ignored. (Page 11, Line 333-334)

31. Page 9, ln 290: 'Moreover, whether the status of river ice was steady or not also could not be neglected when studying the role of snow cover. ' Not clear what you mean by 'steady' river ice. Please clarify. Also, it is not clear how the role of snow cover is influenced by the steadiness (?) of river ice.

**Reply to comments:** Thanks for the doubt, and we have to admit the usage of "steady" is confusing. When the river is completely frozen, the ice cover grows as the air temperature becomes lower and lower. The ice cover wouldn't melt until next spring, and the status is steady. That's what we planned to express, and in the new manuscript, "steady" is replaced by "completely frozen ". (Page 11, Line 350-351)

32. Page 9, ln 295: 'and the snow mixes with surface ice into slush and promotes melting.' Please elaborate this a bit more as snow can turn into slush regardless of the presence of ice. Also, please briefly elaborate (add a sentence) on why would the slush promote melting.

**Reply to comments:** Thank you for this helpful suggestion, and we elaborate this as follows: During the ice-decay process, the lake bottom ice stops to grow, and the surface snow or ice melts, and slush forms. The speed of melting depends on the ability to absorb heat, and the slush can absorb more heat, which promotes melting. The slush often exists through multiple freeze/thaw cycles of river ice before completely disappearing. Therefore, the status of river ice could not be neglected when studying the role of snow cover. (Page 11, Line 346-350)

33. Section 3.3.2. In this section it would be interesting to relate you finding from the

regression modelling and the zones you identifies from EOF analysis. Is the model performing better or worse for different zones.

**Reply to comments:** Thank you for this wonderful idea. In this study, we tried to link regression modeling and the zones, but it didn't work. As we discussed above, the typical geographical zones were identified visually from EOF results of completely frozen duration, and EOF is not a reasonable method for classification. Instead, we used grouping analysis to determine the classification. Comparing the Calinski-Harabasz pseudo F-statistic with group numbers. The completely frozen duration performed were divide into four groups. However, the maximum ice thickness fails to obtain the classification. Yet, there doesn't exist a unified geographical zoning method, which is suitable for both ice phenology and ice thickness pattern analysis.

What's more, the grouping analysis used the average values of complete frozen duration for each hydrological station with dataset size of 156. The complete frozen duration only has one value within one cold season. The regression modeling used the ice thickness measured every five days. The ice thickness had 37 measurements within one cold season, and we selected 31 of 37, ranging from November to March. Only 120 of 156 stations had ice thickness measurements. In a word, complete frozen duration and regression modeling didn't share the same dataset. That's another reason we fail to achieve this idea.

34. Page 9; Ln 302: 'which is consistent with previous studies. ' Provide some references here for the studies.

**Reply to comments:** Thanks for your insightful suggestions. Sorry for the missing reference, and we updated the reference. Figure 9 indicates that snow depth outweighed air temperature in terms of the effect on ice thickness, which is consistent with previous studies (Sharma et al., 2019; Magnuson et al., 2000). (Page 12, Line 366-367)

**Reply to comment:** Thanks for your insightful suggestions. The sentence has been removed. We have updated the last paragraph of the Conclusion.

[revised manuscript text omitted]

---

## Author Response (AR3)

Dear Valentina,

Thank you for considering our manuscript entitled "The role of snow cover on ice regime across Songhua River Basin, Northeast China" (tc-2019-242) for publication with the journal of The Cryosphere. Those comments are all valuable and very helpful for revising and improving our paper. We carefully gone through the comments and made corrections accordingly (these marked as red in the manuscript). The language of our manuscript have been refined and polished by a professional editing company.

Best wishes,
Qian Yang

Title: I think the title is somewhat narrowly defined as you analyse more than snow cover as predictor of ice thickness evolution. I suggest an alternative title: 'Investigation of spatial and temporal variability of river-ice phenology and thickness across Songhua River Basin, Northeast China'

**Reply to comments: Thank you for this helpful suggestion, and we modified as you suggested, seen in Line 1-3.**

Abstract:

'ice phenology and ice thickness...' change to river-ice phenology and river-ice thickness'

**Reply to comments: We amended the relevant parts in the manuscript in accordance with your advices, seen in Line 15-16.**

'Using ice records of local hydrological stations...' Change to -> Using daily ice records of XX hydrological stations across the region, we examined the spatial variability in the ice phenology and ice thickness from 2010 to 2015.'

**Reply to comments: Thanks for your comments and we have revised the manuscript according to your comments, seen in Line 15-16.**

'The cluster pattern of yearly maximum ice thickness has been measured by Global and local Moran's I.' Change to: 'We identified four spatial clusters based on Moran's I spatial autocorrelation applied to yearly maximum ice thickness.'

**Reply to comments: Thank you for this helpful suggestion, and we modified as you suggested, seen in Line 18-21.**

The high values clustered in the Xiao Higgan Mountains...' change to 'High values of ice thickness clustered in the ...

**Reply to comments: We adopted your advice, and modified the relevant parts in manuscript, seen in Line 18-21.**

'Six Bayesian regression models were built between ice thickness, air temperature, and snow depth in three sub-basins of the Songhua River Basin.' Change to: 'In three sub-basins of the Songhua River Basin, we developed six Bayesian regression model to predict ice thickness from air temperature and snow depth.

**Reply to comments: Thanks for your comments and we have revised according to your comments, seen in Line 24-27.**

'The determine R2 of Bayesian linear 25 regression ranged from 0.80 to 0.95, and root mean square errors ranged from 0.08 to 0.18.' Change to: 'The goodness of the fit ($R^2$) for these regression models ranged from 0.80 to 0.95, and the root mean square errors ranged from 0.08 to 0.18' -> shouldn't there be a unit for the root mean square error? Is it meters?

**Reply to comments: Thank you for this helpful suggestion, and we modified as you suggest, seen in Line 26-27 .**

Line 81: 'The reliance on information makes the physically-based model more suitable for small watershed applications within 100 km2 . The empirical model enables it possible to predict the changes in ice regime from limited climate data for larger basin applications (Yang et al., 2020). 'Change to: 'As this information is more

readily available on local scales, the physically-based models are more suitable for small watershed applications (e.g. within 100 km^2). On the other hand, empirical models are more commonly used to predict changes in ice regime from relatively limited climate data available over larger basins (Yang et al, 2020).'

**Reply to comments: I am very grateful to your comments. According with your advice, we amended the relevant part in manuscript, seen in Line 82-86.**

Line 108: 'and compared' -> 'and compare'

**Reply to comments: Thanks for your comments, we have revised and you can check in Line 111-112.**

Line 110: 'was quantitatively explored' -> 'is quantitatively explored'

**Reply to comments: We modified as you suggested, seen in Line 113.**

Line 160-165: Change to paragraph to the following (please correct/edit if necessary):

'Our overall method can be summarized in the following steps: First, we use Kriging to spatially interpolate in situ measurements of ice phenology. Second, we use Morai's I spatial autocorrelation to identify spatial clusters based on the interpolated ice phenology data. Finally, for each cluster, we analyze the drivers of spatial and temporal variability of the river ice thickness. To do so, we use the Bayesian linear regression to quantify the links between the river ice thickness and snow depth and air temperature.'

**Reply to comments: I am very grateful to your comments. According with your advice, we amended the relevant part in manuscript, please check the revised manuscript for details (Line 163-169).**

Line 370-373: Change this section to:

'For the Bayesian linear regressions, we used the field measurements that span from November to March, thus focusing only on the cold part of the year. During this period, the river surface is completely frozen, and the air temperature that falls below

0℃ promotes the ice growth. April is the month when the rise of air temperatures above 0℃ enables the river ice to melt.'

**Reply to comments: Thanks for your comments and we have revised according to your comments, seen in Line 375-379.**

Line 375-380: Change this section to:

'The correlation in Figure 7 between air temperature and ice regime was not as significant as found in some previous studies (e.g. Gao and Stefan, 2004). One of the reasons is that previous studies often averaged the air temperatures over a longer period and at a regional scale, therefore loosing the signal on seasonality at a local scale (e.g. Pavelsky and Smith, 2004; Yang et al., 2020). To circumvent this shortcoming, we applied the regression analysis on seasonal time series of ice thickness and air temperature.'

**Reply to comments: Thank you for this helpful suggestion, and we modified as you suggest, seen in Line 382-388.**

Line 410: 'important role as the ice cover becomes completely frozen' -> shouldn't it be 'as the river becomes completely frozen'?

**Reply to comments: I am very grateful to your comments. According with your advice, we amended the relevant part in manuscript, seen in Line 398.**

Line 414: ' considering two types of air temperature' -> 'considering air temperature, as well as cumulative air temperature.'

**Reply to comments: Thanks for your comments and we have revised according to your comments, seen in Line 427.**

Line 416-417: Change to: 'According to the performance metrics ($R^2$, root mean square error), the cumulative air temperature of freezing is shown to be a better predictor than the air temperature in simulating the ice thickness changes.

**Reply to comments: Thank you for this helpful suggestion, and we modified as you suggest, seen in Line 429-432.**

Line 418: 'ice process' -> 'ice thickness evolution'

**Reply to comments: I am very grateful to your comments. Considering your advice, we have updated the expression, seen in Line 431-433.**

Line 420: Remove this sentence since it is a repetition: 'The results suggested that heat exchanges between the river surface and the atmosphere dominated the ice process, and cumulative air temperature of freezing influenced the thickness is more sensitive indicators of heat loss of ice growth and decay than the air temperature.'

**Reply to comments: Thanks for your comments and we have removed according to your comments.**

Line 425: 'The work herein will provide a valuable reference for the retrieval of ice development by remote sensing. Therefore, we plan to use satellite data to enlarge our study scope in our future work.' Change to: 'Data analysed in this study present a valuable reference for future studies that rely on remote sensing observations of river ice thickness in this area.

**Reply to comments: Thank you for this helpful suggestion, and we modified as you suggested, seen in Line 437-440.**

**CERTIFICATE OF ENGLISH EDITING**

This is to certify that the manuscript entitled **Investigation of spatial and temporal variability of river-ice phenology and thickness across Songhua River Basin, Northeast China** commissioned to us has been carefully edited by two native English-speaking editors of Language Essentials Editing Service, and the grammar, spelling, and punctuation have been verified and corrected where needed. Based on this review, we believe that the language in this paper meets academic journal requirements. Please contact us with any questions.

*Jinmu Yang*

Dr. JinmuYang

Founder of Language Essentials Editing Service

Date of Issue

August 31, 2020

Language Essentials Editing Service is a professional English editing service provide by Qingdao Academically Created World Education Technology Co., Ltd. Our company provides professional English editing and publishing services for scholars all over the world, whose editors come from more than 100 universities and research institutes.
* * *
Qingdao Academically Created World Education Technology Co., Ltd.

College of Environmental Science and Engineering, Ocean University of China, 238 Songling Road, Laoshan District, Qingdao, Shandong Province, China

Contact us: **jinmuacademic@aliyun.com**

[revised manuscript text omitted]